# An Exponential Separation Between Quantum and Quantum-Inspired Classical Algorithms for Linear Systems

Allan Grønlund [* 1]   Kasper Green Larsen [* 2]

## Abstract

Achieving a provable exponential quantum speedup for an important machine learning task has been a central research goal since the seminal HHL quantum algorithm for solving linear systems and the subsequent quantum recommender systems algorithm by Kerenidis and Prakash. These algorithms were initially believed to be strong candidates for exponential speedups, but a lower bound ruling out similar classical improvements remained absent. In breakthrough work by Tang, it was demonstrated that this lack of progress in classical lower bounds was for good reasons. Concretely, she gave a classical counterpart of the quantum recommender systems algorithm, reducing the quantum advantage to a mere polynomial. Her approach is quite general and was named *quantum-inspired classical* algorithms. Since then, almost all the initially exponential quantum machine learning speedups have been reduced to polynomial via new quantum-inspired classical algorithms. From the current state-of-affairs, it is unclear whether we can hope for exponential quantum speedups for any natural machine learning task. In this work, we present the first such provable exponential separation between quantum and quantum-inspired classical algorithms for the basic problem of solving a linear system when the input matrix is well-conditioned and has sparse rows and columns.

## 1. Introduction

Demonstrating an exponential quantum advantage for a relevant machine learning task has been an important research goal since the promising quantum algorithm by Harrow et al.

(2009) for solving linear systems. Ignoring a few details, the HHL algorithm (and later improvements (Ambainis, 2012; Childs et al., 2017)) generates a quantum state $\sum_{i=1}^{n} x_i|i\rangle$ corresponding to the solution $x = M^{-1}y$ to an $n \times n$ linear system of equations $Mx = y$ in just $\text{poly}(\ln n)$ time. At first sight, this seems exponentially faster than any classic algorithm, which probably has to read the entire input matrix $M$ to solve the same problem. However, as pointed out e.g. by Aaronson (2015), the analysis of the HHL algorithm assumes the input matrix is given in a carefully chosen input format. Taking this *state preparation* into consideration, it was initially unclear how the performance could be compared to a classical algorithm and whether any quantum advantage remained.

The shortcoming of the HHL algorithm regarding state preparation was later addressed in several works, with one of the first and most thorough treatments in the thesis of Prakash (2014). Prakash introduced a framework where input matrices and vectors to a linear algebraic machine learning problem are given as simple classical data structures, but with quantum access to the memory representations. This allows for a direct comparison to classical data structures where the input is given in the same data structure. Expanding on these ideas, Kerenidis & Prakash (2017) presented a quantum recommender systems algorithm that was exponentially faster than the best classical counterpart. Their algorithm was one of the strongest candidates for a provable exponential quantum advantage and sparked a fruitful line of research, yielding exponential speedups for a host of important machine learning tasks, including solving linear systems (Chakraborty et al., 2019), linear regression (Chakraborty et al., 2019), PCA (Chakraborty et al., 2019), recommender systems (Kerenidis & Prakash, 2017), supervised clustering (Lloyd et al., 2013) and Hamiltonian simulation (Gilyén et al., 2019).

Despite the exponential speedups over classical algorithms, a lower bound for classical algorithms ruling out a similar improvement via new algorithmic ideas remained elusive. It turned out that this was for good reasons: In breakthrough work by Tang (2019), it was demonstrated that on all inputs where the recommender systems algorithm by Kerenidis and Prakash yielded an exponential speedup, a similar speedup

[1]Kvantify, Aarhus, Denmark [2]Aarhus University, Aarhus, Denmark. Correspondence to: Allan Grønlund <ag@kvantify.dk>, Kasper Green Larsen <larsen@cs.au.dk>.

*Proceedings of the 43rd International Conference on Machine Learning*, Seoul, South Korea. PMLR 306, 2026. Copyright 2026 by the author(s).

could be obtained via a classical algorithmic approach that she dubbed *quantum-inspired classical* (QIC) algorithms. Since then, almost all the initially exponential speedups from quantum algorithms have been reduced to mere polynomial speedups through the development of new efficient QIC algorithms, see e.g. (Bakshi & Tang, 2024; Chia et al., 2022; Shao & Montanaro, 2022). The disheartening state-of-affairs is thus that only a few machine learning problems remain where there is still an exponential gap between quantum and QIC algorithms. Based on Tang's work, it remains entirely plausible that new QIC algorithms may close these gaps as well.

**Our Contribution.** In this work, we present the first provable exponential separation between quantum and quantum-inspired classical algorithms for the central problem of solving linear systems with sparse rows and columns. The lower bound is exponentially higher than known quantum upper bounds (Costa et al., 2022) when the matrix is well-conditioned, thus establishing the separation.

### 1.1. Quantum-Inspired Classical Algorithms

In the following, we formally introduce QIC algorithms, the linear system problem, our lower bound statement and previous work on proving separations between quantum and QIC algorithms.

As mentioned earlier, the work by Prakash (2014), and later work by Kerenidis & Prakash (2017), gave rigorous frameworks for directly comparing a quantum algorithm for a machine learning task with a classical counterpart. Taking state preparation into account, Kerenidis and Prakash define a natural input format (data structure) for matrices and vectors in linear algebraic problems. At a high level, they assume the input is presented as a classical binary tree based data structure over the entries of the rows and columns of a matrix. They then built their quantum recommender system algorithm assuming quantum access to the memory representation of this classical data structure. Follow-up works have used essentially the same input representation or equivalent formulations. In many cases, for sufficiently well-conditioned matrices, the obtained quantum algorithms run in just $\text{poly}(\ln n)$ time.

Now to prove a separation between quantum and classical algorithms, any fair comparison should use the same input representation. Given the simplicity of the data structure by Kerenidis and Prakash for representing the input, it seemed reasonable to conjecture that any classical algorithm for e.g. recommender systems would need polynomial time even when given this data structure. This intuition was however proven false by Tang (2019). Her key insight was that the classical data structure allows efficient classical (i.e. $\text{poly}(\ln n)$ time) $\ell_2$ sampling (formally defined below) from

the rows and columns of the input, as well as efficient reading of individual entries. Exploiting this sampling access, she gave a classical algorithm for recommender systems that runs in just $\text{poly}(\ln n)$ time on all matrices where the quantum algorithm by Kerenidis and Prakash does. She referred to such classical algorithms with $\ell_2$ sampling access to input matrices and vectors as *quantum-inspired classical* algorithms. This sampling access has since then proved extremely useful in other machine learning tasks, see e.g. (Bakshi & Tang, 2024; Chia et al., 2022; Shao & Montanaro, 2022). Tang (2019) summarized the above discussion as follows: "*when quantum machine learning algorithms are compared to classical machine learning algorithms in the context of finding speedups, any state preparation assumptions in the quantum machine learning model should be matched with $\ell_2$-norm sampling assumptions in the classical machine learning model*".

Using the notation of Mande & Shao (2024), QIC algorithms formally have the following access to the input:

**Definition 1.1** (Query Access). *For a vector $v \in \mathbb{R}^n$, we have $Q(v)$, query access to $v$, if for all $i$, we can query $v_i$. Likewise for a matrix $M \in \mathbb{R}^{m \times n}$, we have query access to $M$ if for all $(i,j) \in [m] \times [n]$, we can query $M_{i,j}$.*

**Definition 1.2** (Sampling and Query Access to a Vector). *For a vector $v \in \mathbb{R}^n$, we have $SQ(v)$, sampling and query access to $v$, if we can*

- *Query for entries of $v$ as in $Q(v)$.*

- *Obtain independent samples of indices $i \in [n]$, each distributed as $\mathbb{P}[i] = v_i^2 / \|v\|^2$.*

- *Query for $\|v\|$.*

**Definition 1.3** (Sampling and Query Access to a Matrix). *For a matrix $M \in \mathbb{R}^{m \times n}$, we have $SQ(M)$ if we have $SQ(M_{i,\star})$, $SQ(M_{\star,j})$, $SQ(r)$ and $SQ(c)$ for all $i \in m$ and $j \in n$ where $r(M) = (\|M_{1,\star}\|, \ldots, \|M_{m,\star}\|)$ and $c(M) = (\|M_{\star,1}\|, \ldots, \|M_{\star,n}\|)$. Here $M_{i,\star}$ is the $i$'th row of $M$, $M_{\star,j}$ is the $j$'th column, $r(M)$ is the vector of row-norms and $c(M)$ is the vector of column-norms of $M$.*

In the above definitions, and throughtout the paper, we use $\|x\|$ to denote the $\ell_2$ norm $\|x\|_2$. With the input representation defined, we proceed to present the problem of solving a linear system via a QIC algorithm. Here one again needs to be careful for a fair comparison between quantum and QIC algorithms. Concretely, the known quantum algorithms for solving a linear system $Mx = y$ do not output the full solution $x$ (which would take linear time), but instead a quantum state $\sum_i \tilde{x}_i |i\rangle$ for a $\tilde{x}$ approximating the solution $x$. Taking measurements on such a state allows one to sample an index $i$ with probability $\tilde{x}_i^2 / \|\tilde{x}\|^2$. With this in mind, the classical analog of solving a linear system is as follows.

**Problem 1.4** (Linear Systems)**.** *Given $SQ(M)$ and $SQ(y)$ for a symmetric and real matrix $M \in \mathbb{R}^{n \times n}$ of full rank, a vector $y \in \mathbb{R}^n$ and precision $\varepsilon > 0$, the Linear Systems problem is to support sampling an index $i$ with probability $\tilde{x}_i^2 / \|\tilde{x}\|^2$ from a vector $\tilde{x}$ satisfying that $\|\tilde{x} - x\| \le \varepsilon \|x\|$ where $x = M^{-1} y$ is the solution to the linear system of equations $Mx = y$.*

The query complexity of a QIC algorithm for solving a linear system, is the number of queries to $SQ(M)$ and $SQ(y)$ necessary to sample one index $i$ from $\tilde{x}$. We remark that the known QIC algorithms furthermore output the value $\tilde{x}_i$ upon sampling $i$. Since we aim to prove a lower bound, our results are only stronger if we prove it for merely sampling $i$.

**Quantum Benchmark.** To prove our exponential separation, we first present the state-of-the-art performance of quantum algorithms for linear systems. Here we focus on the case where the input matrix $M$ has sparse rows and columns, i.e. every row and column has at most $s$ non-zero entries. The running time of the best known quantum algorithm depends on the condition number of $M$, defined as

$$\kappa = \sigma_{\max} / \sigma_{\min}.$$

Here $\sigma_{\max}$ is the largest singular value of $M$ and $\sigma_{\min}$ is the smallest singular value. Note that for real symmetric $M$ of full rank, all eigenvalues $\lambda_1 \ge \cdots \ge \lambda_n$ of $M$ are real and non-zero, and the singular values $\sigma_{\max} = \sigma_1 \ge \cdots \ge \sigma_n = \sigma_{\min} > 0$ are the absolute values of the eigenvalues $\{|\lambda_i|\}_{i=1}^n$ in sorted order. Given a precision $\varepsilon > 0$, matrix $M$ and vector $y$ as input (in the classical data structure format), the quantum algorithm by Costa et al. (2022) (improving over previous works, e.g. (Chakraborty et al., 2019)) runs in time

$$O(\kappa \ln(1/\varepsilon) \min\{s, \sqrt{s}(\kappa s / \varepsilon)^{o(1)}\}) \qquad (1)$$

to produce a quantum state $\sum_i \tilde{x}_i |i\rangle$ for a $\tilde{x}$ with $\|\tilde{x} - x\| \le \varepsilon \|x\|$ with $x = M^{-1} y$. We remark that to derive (1) from (Chakraborty et al., 2019), one invokes either a reduction from (Gilyén et al., 2019) (for the $s$ guarantee) or (Low, 2019) (for the $\sqrt{s}(\kappa s / \varepsilon)^{o(1)}$ guarantee) to obtain a block-encoding of a sparse matrix. See also the recent work (Low & Su, 2024) for high success probability guarantees.

**QIC Benchmark.** For QIC algorithms, the best bound is due to Shao & Montanaro (2022) and has a query complexity (and running time) of

$$\text{poly}(s, \kappa_F, \ln(1/\varepsilon), \ln n), \qquad (2)$$

where

$$\kappa_F = \|M\|_F / \sigma_{\min} = \frac{\sqrt{\sum_i \sigma_i^2}}{\sigma_{\min}}.$$

Since $\kappa_F$ may be larger than $\kappa$ by as much as a $\sqrt{n}$ factor, there are thus matrices with $\kappa, s = \text{poly}(\ln n)$ where there is an exponential gap between (1) and (2).

An alternative upper bound for a closely related problem was given by Gharibian & Le Gall (2022) and is based on the quantum singular value decomposition. In more detail, they show that for a matrix $M$ where $M^\dagger M$ has eigenvalues $1 = \lambda_1 \ge \cdots \ge \lambda_n = \kappa^{-1}$, if we apply a degree $d$ polynomial $P$ to the square-roots of the eigenvalues (and maintain the same eigenvectors) to obtain a matrix $P(\sqrt{M^\dagger M})$ with eigenvalues $P(\lambda_1^{1/2}), \ldots, P(\lambda_n^{1/2})$, then given sampling and query access to $M$ and two vectors $u$ and $v$ of unit norm, it is possible to classically estimate the number $u^\dagger P(\sqrt{M^\dagger M}) v$ to within additive $\epsilon$ in roughly $O(s^{2d} / \varepsilon^2)$ time. Since $x^{-1}$ can be well-approximated in the interval $[-1, 1] \setminus [-\kappa^{-1}, \kappa^{-1}]$ by a degree $d = O(\kappa \log(\kappa / \varepsilon))$ polynomial, this essentially allows us to estimate $u^\dagger M^{-1} v$. This is not quite sampling from $M^{-1} v$ but reasonably close. Furthermore, it suggests that to achieve an exponential separation between QIC and quantum algorithms, we probably need to consider $\kappa$ that is at least logarithmic in $n$. See the discussion below for a similar recent result by Montanaro and Shao and more details.

In light of the above, it is clear that for many settings of parameters $\kappa, \kappa_F$ and $\varepsilon$, there is still an exponential gap between the quantum and QIC upper bounds. However it has still not been proved unconditionally that such a gap is inherent.

**Our Result.** We show the following strong lower bound for QIC algorithms

**Theorem 1.5.** *There is a constant $c \ge 1$, such that for any integers $n, k \ge c$, it holds for any QIC algorithm $\mathcal{A}$ with precision $\varepsilon \le 2^{-ck}$ for linear systems, that there exists a full rank $n \times n$ symmetric real matrix $M$ with condition number $\kappa \le c \ln n$ and 3-sparse rows and columns, such that $\mathcal{A}$ must make $c^{-1} n^{1-1/k}$ queries to $SQ(M)$ on the linear system $Mx = e_1$.*

Observe that the complexity of the best quantum algorithm (1) for this setting of $s, \kappa$ and $\varepsilon$ is just $\ln n$, hence the claimed exponential separation. Furthermore, the matrix $M$ is extremely sparse, with only $s = 3$ non-zeroes per row and column, and the vector $y$ in the linear system $Mx = y$ is simply the first standard unit vector $e_1$. Finally, the lower bound holds even for a constant precision $\varepsilon$. In particular, for any constant $k$, the precision $\varepsilon$ is only required to be less than a constant and still the number of queries must be $\Omega(n^{1-1/k})$.

Let us also remark that our lower bound also holds in the well-studied *sparse access model*, see e.g. (Gilyén et al., 2019; Dervovic et al., 2018). In the sparse access model, we

can query for the $i$'th non-zero of any row or column of $M$ and all rows and columns have at most $s$ non-zeros.

**Previous Hardness of Solving Linear Systems.** In the seminal HHL paper (Harrow et al., 2009) that introduced the first quantum algorithm for solving linear systems, the authors also proved conditional lower bounds for classical algorithms. Concretely, they proved that solving a sparse and well-conditioned linear system is BQP-complete (bounded-error quantum polynomial time). For this result, the assumption is that the matrix is presented to the algorithm as a short binary encoding (a $\mathrm{poly}(\ln n)$ bit encoding of an $n \times n$ matrix). Assuming the widely-believed conjecture that BQP$\neq$P, this implies that there is no polynomial time classical algorithm for the same problem and input representation. Since QIC access to the matrix can be efficiently simulated from such a binary encoding, this implies a *conditional* separation between quantum and QIC algorithms for linear systems. Our new lower bound is however *unconditional* and gives an explicit gap ($\ln n$ vs. $n^{1-1/k}$ for any constant $k$) in terms of the number of oracle queries without relying on computational hardness assumptions. However, it is reasonable to say that the HHL result hints that an unconditional separation was within reach.

**Previous Lower Bounds for QIC Algorithms.** Prior to our work, there has been several works proving lower bounds for QIC and related algorithms, although none of them establishing the exponential separation for linear systems that we give in this work. In the following, we review the most relevant such works.

Recent work by Mande & Shao (2024) also study linear systems among several other problems. Using reductions from number-in-hand multiparty communication complexity (Phillips et al., 2012), they prove a number of lower bounds for QIC algorithms for linear regression (and systems), supervised clustering, PCA, recommender systems and Hamiltonian simulation. Their lower bounds are of the form $\tilde{\Omega}(\kappa_F^2)$, but only for problems where the best known quantum algorithms are no better than $\tilde{O}(\kappa_F)$, thus establishing quadratic separations compared to our exponential separation. Let us also remark that our lower bound proof takes a completely different approach, instead reducing from a problem of random walks by Childs et al. (2003), or alternatively, from $k$-Forrelation (Aaronson & Ambainis, 2015).

Andoni et al. (2019) also studied the problem of approximating the solution to a linear system in sub-linear time via sampling access to the non-zero entries of $M$ and $y$. However, instead of requiring that the output index is sampled from a vector $\tilde{x}$ with $\|x - \tilde{x}\| \leq \varepsilon\|x\|$, they make the stronger requirement that for any query index $i$, the algorithm must approximate $x_i$ to within additive $\varepsilon\|x\|_\infty$. Note that this is both a harder problem in terms of the goal of approximating every single entry of $x$, and the error guarantee in terms of $\varepsilon\|x\|_\infty$ is much stricter than our $\varepsilon\|x\|$. Proving lower bounds for their problem is thus seemingly an easier task. In particular, their hard instance is trivial to solve with additive $\varepsilon\|x\|$ error for $\varepsilon = n^{-o(1)}$ as one can simply output 0 to approximate $x_i$ for any $i$.

Very recently, Montanaro & Shao (2024) studied the sparse access model for the problem of estimating entries of $f(M)$. Here $f : [-1, 1] \to [-1, 1]$ is a function that can be approximated by a polynomial and $f(M)$ for a Hermitian $M$ with $\|M\| \leq 1$ (all eigenvalues bounded by 1 in absolute value) with eigendecomposition $UDU^\dagger$, is the matrix $Uf(D)U^\dagger$ with $f$ applied entrywise on the diagonal entries of $D$. In particular, for $f(x) = x^{-1}$, we have that $f(M)$ is the inverse $M^{-1}$. Montanaro and Shao show that the quantum query complexity of estimating entries of $f(M)$ is bounded by a polynomial in $\widetilde{\deg}_\varepsilon(f)$, where $\widetilde{\deg}_\varepsilon(f)$ is the least degree of a real-valued polynomial $P$ such that $|P(x) - f(x)| \leq \varepsilon$ for all $x \in [-1, 1]$. On the other hand, they also prove that classical algorithms must query $\exp(\Omega(\widetilde{\deg}_\varepsilon(f)))$ many entries. This result does not directly apply to the matrix inverse problem since $f(x) = x^{-1}$ is not bounded by 1 in absolute value for $x \in [-1, 1]$. Montanaro and Shao discuss this shortcoming and observe that if all eigenvalues of $M$ are at least $\kappa^{-1}$ in absolute value (corresponding to condition number $\kappa$), then $f(x) = x^{-1}$ can be approximated in $[-1, 1] \setminus [-\kappa^{-1}, \kappa^{-1}]$ by a polynomial of degree $O(\kappa \log(\kappa/\varepsilon))$. However, this is an upper bound, not a lower bound on the approximate degree, and they mention that they are not aware of any lower bound on the approximate degree of $f(x) = x^{-1}$ when restricted to $[-1, 1] \setminus [-\kappa^{-1}, \kappa^{-1}]$. An appropriate rescaling of $M$ together with this upper bound result thus hints that the condition number $\kappa$ must be at least logarithmic in the matrix size to establish an exponential separation for estimating entries of $M^{-1}$ to within additive $\varepsilon$. On the other hand, for general $f$ with approximate degree at least logarithmic, their results give exponential separations between classical and quantum algorithms for estimating entries of $f(M)$ in the sparse access model.

## 2. Separation

We prove our lower bound result in Theorem 1.5 via a reduction from either a problem by Childs et al. (2003) on random walks in graphs, or from the $k$-Forrelation problem (Aaronson & Ambainis, 2015). While the reduction from $k$-Forrelation gives the tightest lower bound, we believe both reductions have value. In particular, we find that the reduction from random walks is simpler and more self-contained. Furthermore, the proof via random walks provides novel new insights into the graph construction by Childs et al. (2003). We hope these insights may find

further applications and have therefore chosen to include both reductions. We start by presenting our reduction from random walks.

Childs et al. (2003) study the following oracle query problem. There is an unknown input graph. The graph is obtained by constructing two perfect binary trees $T_1$ and $T_2$ of height $n$ each, i.e. they have $2^n$ leaves. The leaves of the two trees are connected by a uniformly at random chosen alternating cycle. That is, if we fix an arbitrary leaf $\ell_1$ in tree $T_1$, then the cycle is obtained by connecting $\ell_1$ to a uniform random leaf in $T_2$, that leaf is then again connected to a uniform random remaining leaf in $T_1$ and so forth, always alternating between the two trees. When no more leaves remain and we are at leaf $\ell_2$ in $T_2$, the cycle is completed by adding the edge back to $\ell_1$.

The $N = 2^{n+2} - 2$ nodes of $T_1$ and $T_2$ are now assigned uniform random and distinct $2n$ bit labels, except the root of $T_1$ that is assigned the all-0 label. Call the resulting random graph $G_n$.

Childs et al. now consider the following game: We are given query access to an oracle $\mathcal{O}$. Upon receiving a $2n$-bit query string $x$, $\mathcal{O}$ either returns that no node of the two trees has the label $x$, or if such a node exists, the labels of its neighbors are returned. The game is won if $\mathcal{O}$ is queried with the label of the root of $T_2$. Otherwise it is lost. Throughout the paper, we use $i^\star$ to denote the (random) $2n$-bit label of the root of $T_2$. Childs et al. (2003) prove the following lower bound for any classical algorithm that accesses the graph $G_n$ only through the oracle $\mathcal{O}$:

**Theorem 2.1** (Childs et al. (2003)). *Any classical algorithm that queries $\mathcal{O}$ at most $2^{n/6}$ times wins with probability at most $4 \cdot 2^{-n/6}$.*

The key idea in our lower bound proof is to use an efficient QIC algorithm for linear systems to obtain an efficient classical algorithm for the above random walk game. As a technical remark, Childs et al. also included a random *color* on each edge of $G_n$ and this color was also returned by $\mathcal{O}$. The colors were exploited in a quantum upper bound making only $\text{poly}(n)$ queries to the oracle to win the game with good probability. Since providing an algorithm with less information only makes the problem harder, the lower bound in Theorem 2.1 clearly holds for our variant without colors as well.

## 2.1. Reduction from Random Walks

Assume we have a QIC algorithm $\mathcal{A}$ for linear systems that makes $T(M, \varepsilon)$ queries to $SQ(M)$ to sample an index from $\tilde{x}$ such that each index $i$ is sampled with probability $\tilde{x}_i^2 / \|\tilde{x}\|^2$ for a $\tilde{x}$ satisfying $\|\tilde{x} - x\| \leq \varepsilon \|x\|$ for a sufficiently small $\varepsilon > 0$. Here $x = M^{-1}y$ is the solution to the equation $Mx = y$. Our idea is to carefully choose a matrix $M$ (and

vector $y$) depending on the random graph $G_n$ represented by $\mathcal{O}$, such that simulating all calls of $\mathcal{A}$ to $SQ(M)$ and $SQ(y)$ via calls to the oracle $\mathcal{O}$, $\mathcal{A}$ will output a sample index $i$ that with sufficiently large probability equals the index $i^\star$ of the root of $T_2$. Said differently, the coordinate $\tilde{x}_{i^\star}$ of $\tilde{x}$ corresponding to the root $i^\star$ of $T_2$ is sufficiently large. Making a final query to $\mathcal{O}$ with $i$ then wins the game with good probability.

Let us first describe the simulation and the matrix $M$. We consider the $2^{2n} \times 2^{2n}$ matrix $M$ whose rows and columns correspond to the possible $2n$-bit labels in the graph $G_n$. Let $B$ be the $N \times N$ submatrix of $M$ corresponding to the (random) labels of the $N = 2^{n+2} - 2$ actual nodes in $G_n$. If $A$ denotes the adjacency matrix of the graph $G_n$ (nodes ordered as in $B$), then we let $B = \lambda I - A$ for a parameter $\lambda$ to be determined. For all rows/columns of $M$ not corresponding to nodes in $G_n$, we let the diagonal entry be $\sqrt{\lambda^2 + 3}$ and all off-diagonals be 0. Note that we are not explicitly computing the matrix $M$ to begin with. Instead, we merely argue that by querying $\mathcal{O}$, we can simulate $\mathcal{A}$ as if given access to $SQ(M)$.

Clearly $M$ is symmetric and real and thus has real eigenvalues and eigenvectors. Furthermore, we will choose $\lambda$ such that $M$ has full rank (which is if and only if all its eigenvalues are non-zero). Also observe that every row/column not corresponding to the two roots of $T_1$ and $T_2$, has norm $\sqrt{\lambda^2 + 3}$ (every node of $T_1$ and $T_2$, except the roots, have degree 3).

Our goal is now to use the QIC algorithm $\mathcal{A}$ for linear systems with $SQ(M)$ and $SQ(e_{0\cdots 0})$ as input, to give an algorithm for the oracle query game with oracle $\mathcal{O}$ for $G_n$. Here $e_{0\cdots 0}$ is the standard unit vector corresponding to the root node of $T_1$ which always has the all-0 label. For short, we will assume this is the first row/column of $M$ and write $e_1 = e_{0\cdots 0}$.

**Simulating $SQ(M)$.** We now argue how to simulate each of the queries on $SQ(M)$ and $SQ(e_1)$ made by $\mathcal{A}$ via calls to $\mathcal{O}$. $SQ(e_1)$ is trivial to implement as $e_1$ is known and requires no calls to $\mathcal{O}$.

For $SQ(M)$, to sample and index from a row $M_{i,\star}$ or column $M_{\star,j}$, we query $\mathcal{O}$ for all nodes adjacent to $i$ (or $j$). If $\mathcal{O}$ returns that $i$ is not a valid label, we return the entry $i$ (corresponding to sampling the only non-zero entry of $M_{i,\star}$, namely the diagonal). Otherwise, $\mathcal{O}$ returns the list of neighbors of $i$. We know that row $i$ (column $j$) has $\lambda$ in the diagonal and $-1$ in all entries corresponding to neighbors. We can thus sample from $M_{i,\star}$ with the correct distribution and pass the result to $\mathcal{A}$. To query an entry $M_{i,j}$ of $M$, we simply query $\mathcal{O}$ for the node $i$. If $\mathcal{O}$ returns that $i$ is not a valid label, we return $\sqrt{\lambda^2 + 3}$ if $j = i$ and otherwise return 0. If $\mathcal{O}$ returns the neighbors of $i$, then if $j = i$, we return $\lambda$,

if $j$ is among the neighbors, we return $-1$ and otherwise we return 0.

We support sampling from $r(M)$ and $c(M)$ by returning a uniform random index among the $2^{2n}$ rows/columns. This requires no queries to $\mathcal{O}$. Note that this *almost* samples from the correct distribution: Every node $v$ of $G_n$ that is not a root of the two trees, is adjacent to three nodes. Hence the norm of the corresponding row and column is precisely $\sqrt{\lambda^2 + 3}$. This is the same norm as all rows and columns of $M$ not corresponding to $G_n$ (i.e. rows and columns outside the submatrix $B$). Hence the distribution of a sample from $r(M)$ and $c(M)$ is uniform random if we condition on not sampling an index corresponding to a root. Conditioned on never sampling a root in our simulation, the simulation of all samples from $r(M)$ and $c(M)$ follow the same distribution as if we used the correct sampling probabilities for $r(M)$ and $c(M)$. The probability we sample one of the two roots is at most $2T(M, \varepsilon)2^{-2n}$ and we subtract this from the success probability of the simulation.

Finally, when the simulation of $\mathcal{A}$ terminates with an output index $i$, we query $\mathcal{O}$ for $i$ (with the hope that $i$ equals the index $i^\star$ of the root of $T_2$). We thus have

**Observation 2.2.** *Given a precision $\varepsilon > 0$ and a QIC algorithm $\mathcal{A}$ for linear systems making $T(M, \varepsilon)$ queries to $SQ(M)$, there is a classical algorithm for random walks in two binary trees that makes at most $T(M, \varepsilon) + 1$ queries to the oracle $\mathcal{O}$, where $M$ is the matrix defined from $G_n$. Furthermore, it wins the game with probability at least $\tilde{x}_{i^\star}^2 / \|\tilde{x}\|^2 - 2T(M, \varepsilon)2^{-2n}$ for a $\tilde{x}$ satisfying $\|\tilde{x} - x\| \leq \varepsilon\|x\|$ with $x = M^{-1}e_1$ the solution to the linear system $Mx = e_1$ and $i^\star$ the index of the root of $T_2$.*

What remains is thus to determine an appropriate $\lambda$, to argue that $M$ has a small condition number, to find a suitable $\varepsilon > 0$ and to show that $\tilde{x}_{i^\star}^2 / \|\tilde{x}\|^2$ is large. This is done via the following two auxiliary results

**Lemma 2.3.** *The adjacency matrix $A$ of the graph $G_n$ has no eigenvalues in the range $(\sqrt{8}, 3 - 2^{-n}]$.*

**Lemma 2.4.** *If $\lambda$ is chosen as $\sqrt{8} + \gamma$ for a $0 < \gamma \leq \left(\frac{1}{16(n+2)}\right)^2$ and $n \geq c$ for a sufficiently large constant $c > 0$, then $x_{i^\star}^2 = \Omega(n^{-5}\|x\|^2)$ where $x = M^{-1}e_1$ is the solution to the linear system $Mx = e_1$ and $i^\star$ is the index of the root of $T_2$.*

Before proving these results and motivating the bounds they claim, we use them to complete our reduction and derive our lower bound. In light of Lemma 2.4, we choose $\lambda = \sqrt{8} + \gamma$ with $\gamma = 1/(16(n+2))^2$. Let us now analyse the condition number of $M$. First, note that the sum of absolute values in any row or column of $M$ is no more than $\max\{\sqrt{\lambda^2 + 3}, \lambda + 3\} \leq 6$. Thus the largest singular value of $M$ is at most 6. For the smallest singular value, observe that $M$ is block-

diagonal with $B$ in one block and $\sqrt{\lambda^2 + 3} \cdot I$ in the other. The latter has all singular values $\sqrt{\lambda^2 + 3} > 3$. For $B$, we first observe that any row and column of the adjacency matrix $A$ has sum of absolute values at most 3. Hence the eigenvalues of $A$ lie in the range $[-3, 3]$. From Lemma 2.3, we now have that all eigenvalues of $B = \lambda I - A$ lie in the ranges $[\lambda - 3, \lambda - 3 + 2^{-n}] \subseteq [-0.18, -0.17]$ (for $n$ sufficiently large) and $(\lambda - \sqrt{8}, \lambda + 3] \subseteq (\gamma, 6)$. We thus have that the smallest singular value (smallest absolute value of an eigenvalue) of $M$ is at least $\gamma = 1/(16(n+2))^2$. The condition number $\kappa$ is thus at most $6 \cdot (16(n+2))^2$ (recall that the size of the matrix is $2^{2n}$, thus the condition number is only polylogarithmic in the matrix size).

Let us next analyse $\tilde{x}$. From Lemma 2.4 we have that $|\tilde{x}_{i^\star}| \geq |x_{i^\star}| - \varepsilon\|x\| = |x_{i^\star}| - O(\varepsilon n^{2.5}|x_{i^\star}|)$. For $\varepsilon \leq (cn^{2.5})^{-1}$ for sufficiently large constant $c$, this is at least $|x_{i^\star}|/2 = \Omega(n^{-2.5}\|x\|)$ by Lemma 2.4. Furthermore, by the triangle inequality, we have $\|\tilde{x}\| \leq (1 + \varepsilon)\|x\|$, and hence $\tilde{x}_{i^\star}^2 / \|\tilde{x}\|^2 = \Omega(n^{-5})$. We thus win the random walk game with probability at least

$$\Omega(n^{-5}) - 2T(M, \varepsilon)2^{-2n}.$$

For $n$ sufficiently large, this implies that either $T(M, \varepsilon) \geq 2^{n/6}$ or this success probability is greater than $4 \cdot 2^{-n/6}$. Thus Theorem 2.1 and Observation 2.2 gives us that the number of queries, $T(M, \varepsilon) + 1$, must be at least $2^{n/6}$. This establishes Theorem 1.5 with slightly worse parameters as the size of $M$ is $2^{2n} \times 2^{2n}$ (i.e. with $\kappa \leq c \ln^2 n$, $\varepsilon \leq (c \ln^{2.5} n)^{-1}$, 4-sparse rows and columns and $\Omega(n^{1/12})$ queries on an $n \times n$ matrix). Our later reduction from $k$-Forrelation (in Section A), improves these parameters by polynomial factors (and $\varepsilon$ needs only be less than a constant).

**Motivating the Construction.** Before we proceed to give proofs in Section B, let us make a number of comments on the intuition behind our concrete choice of matrix $M$ and the bounds claimed in Lemma 2.3 and Lemma 2.4. In light of the reduction from the random walk problem, it is clear that we should choose a matrix $M$ such that we can 1.) simulate $SQ(M)$ via $\mathcal{O}$, 2.) guarantee a small condition number of $M$, and 3.) guarantee that $\tilde{x}_{i^\star}^2 / \|\tilde{x}\|^2$ is as large as possible.

Condition 1. naturally hints at using the adjacency matrix $A$ of $G_n$ as part of the construction of $M$ since querying $\mathcal{O}$ precisely retrieves the neighbors and thus non-zero entries of a row or column of $A$. It further guarantees sparse rows and columns in $M$. For condition 2., we have to introduce something in addition to $A$ as the eigenvalues of $A$ lie in the range $[-3, 3]$, but we have no guarantee that they are sufficiently bounded away from 0. A natural choice is $B = \lambda I - A$ as this shifts the eigenvalues to lie in the range $[\lambda - 3, \lambda + 3]$. Setting the entries of $M$ outside the submatrix

corresponding to the actual nodes in $G_n$ to $\sqrt{\lambda^2 + 3}$ is again a natural choice as this makes sampling from $c(M)$ and $r(M)$ trivial. From this alone, it would seem that $\lambda \gg 3$ would be a good choice. Unfortunately, such a choice of $\lambda$ would not guarantee condition 3. To get an intuition for why this is the case, recall that $\tilde{x}$ is $\varepsilon$-close to $x = M^{-1}e_1$. Since $M$ is block-diagonal, with $B$ the block containing the first row and $\sqrt{\lambda^2 + 3} \cdot I$ the other block, $x$ is non-zero only on coordinates corresponding to the block $B$. Abusing notation, we thus write $x = B^{-1}e_1$. Thus what we are really interested in showing, is that $x_{i^\star}^2 / \|x\|^2$ is large for this $x$. Examining the linear system $Bx = e_1$, we see that all rows corresponding to internal nodes $v$ of $T_1$ and $T_2$ define an equality. The equality states that the entry $x_v$ corresponding to $v$ must satisfy $\lambda x_v - x_{p(v)} - x_{\ell(v)} - x_{r(v)} = 0$ where $p(v)$ is the parent of $v$, $\ell(v)$ the left child and $r(v)$ the right child (for leaves of $T_1$ and $T_2$, the children are the neighboring leaves in the opposite tree). Since this pattern is symmetric across the nodes of $T_1$ and $T_2$, it is not surprising that $x$ is such that all nodes $v$ on the $j$'th level of $T_i$ have the same value $x_v = \psi_j^{(i)}$. Let us focus on the tree $T_2$ and simplify the notation by dropping the index and letting $\psi_j \leftarrow \psi_j^{(2)}$. From the above, the $\psi_j$'s satisfy the constraints $\lambda \psi_{j+1} - \psi_j - 2\psi_{j+2} = 0$. A recurrence $a\psi_{j+2} + b\psi_{j+1} + c\psi_j = 0$ is known as a *second degree linear recurrence* and when $b^2 > 4ac$ its solutions are of the form

$$\psi_j = \alpha \cdot \left( \frac{-b + \sqrt{b^2 - 4ac}}{2a} \right)^j + \beta \cdot \left( \frac{-b - \sqrt{b^2 - 4ac}}{2a} \right)^j,$$

where $\alpha, \beta \in \mathbb{R}$ are such that $\psi_1$ and $\psi_2$ satisfy any initial conditions one might require. For our construction, we have $a = -2, b = \lambda$ and $c = -1$, resulting in

$$\psi_j = \alpha \cdot \left( \frac{\lambda - \sqrt{\lambda^2 - 8}}{4} \right)^j + \beta \cdot \left( \frac{\lambda + \sqrt{\lambda^2 - 8}}{4} \right)^j.$$

Since the root of $T_2$ does not have a parent, the corresponding equality from $Mx = e_1$ gives the initial condition $\lambda \psi_1 - 2\psi_2 = 0$, implying $\psi_1 = (2/\lambda)\psi_2$. If we work out the details, this can be shown to imply $\alpha = -\beta$ in any solution to the recurrence and thus

$$\psi_j = \alpha \cdot \left( \left( \frac{\lambda - \sqrt{\lambda^2 - 8}}{4} \right)^j - \left( \frac{\lambda + \sqrt{\lambda^2 - 8}}{4} \right)^j \right).$$

For $\lambda$ sufficiently larger than $\sqrt{8}$, the term $((\lambda + \sqrt{\lambda^2 - 8})/4)^j$ is at least a constant factor larger in absolute value than $((\lambda - \sqrt{\lambda^2 - 8})/4)^j$ and we have

$$\psi_j \approx -\alpha \cdot \left( \frac{\lambda + \sqrt{\lambda^2 - 8}}{4} \right)^j. \tag{3}$$

For $\lambda \geq 3$, we therefore roughly have that $|\psi_j| \geq |\psi_{j-1}| \cdot (3 + \sqrt{9 - 8})/4 = |\psi_{j-1}|$. But there are $2^j$ nodes of depth $j$ in $T_2$, thus implying that the magnitude of $x_{i^\star}^2$, with $i^\star$ the root of $T_2$, is no more than $\|x\|^2 2^{-n}$. This is far too small for both the sampling probability $x_{i^\star}^2 / \|x\|^2$ and the required precision $\varepsilon$ such that simply setting $\tilde{x}_{i^\star} \leftarrow 0$ in $\tilde{x}$ does not violate $\|\tilde{x} - x\| \leq \varepsilon \|x\|$.

To remedy this, consider again (3) and let us understand for which values of $\lambda$ that $x_{i^\star}^2$ is not significantly smaller than $\|x\|^2$. If we could choose $\lambda = \sqrt{8}$ (technically we require $b^2 > 4ac$ and thus $\lambda > \sqrt{8}$), then (3) instead gives $|\psi_j| \approx |\psi_{j-1}| \cdot \sqrt{8}/4$ and thus $\|x\|^2 \approx \sum_{j=0}^{n-1} 2^j (\sqrt{8}/4)^{2j} x_{i^\star}^2 = nx_{i^\star}^2$. This is much better as the size of the matrix $M$ is $2^{2n}$ and thus an $\varepsilon$ that is only inverse polylogarithmic in the matrix size suffices to sample $i^\star$ with large enough probability to violate the lower bound for the random walk game. Notice also that choosing $\lambda = \sqrt{8} + c$ for any constant $c > 0$ is insufficient to cancel the exponential growth in number of nodes of depth $j$ (as for $\lambda \geq 3$). Thus we are forced to choose $\lambda$ very close to $\sqrt{8}$.

Unfortunately choosing $\lambda \approx \sqrt{8}$ poses other problems. Concretely the eigenvalues of $B$ then lie in the range $[\sqrt{8} - 3, \sqrt{8} + 3]$, which contains 0 and thus we are back at having no guarantee on the condition number. However even if $A$ has eigenvalues in the range $[-3, 3]$, it is not given that there are eigenvalues spread over the entire range. In particular, since all nodes of $G_n$, except the roots, have the same degree 3, we can almost think of $G_n$ as a 3-regular graph. For $d$-regular graphs, the best we can hope for is that the graph is Ramanujan, i.e. all the eigenvalues of the adjacency matrix $A$, except the largest, are bounded by $2\sqrt{d - 1}$ in absolute value (the second largest eigenvalue for any $d$-regular graph is at least $2\sqrt{d - 1} - o(1)$, see (Alon, 1991)). For $d = 3$, this is precisely $\sqrt{8}$. In our case, the graph $G_n$ is sadly *not* Ramanujan, as in addition to the largest eigenvalue of roughly 3 (only roughly since our graph is not exactly 3-regular), it also has an eigenvalue of roughly $-3$. However, in Lemma 2.3 we basically show that this is the *only* other eigenvalue that does not lie in the range $[-\sqrt{8}, \sqrt{8}]$.

In light of the above, we wish to choose $\lambda$ as close to $\sqrt{8}$ as possible, but not quite $\lambda = \sqrt{8}$ as we may then again risk having an eigenvalue arbitrarily close to 0. If we instead choose $\lambda = \sqrt{8} + \gamma$ for a small $\gamma > 0$, then (3) gives us something along the lines of

$$|\psi_j| \approx |\psi_{j-1}| \cdot \left( \frac{\sqrt{8} + \sqrt{8 + 2\sqrt{8}\gamma + \gamma^2 - 8}}{4} \right)$$

$$= |\psi_{j-1}| \cdot \frac{\sqrt{8}}{4} \cdot (1 + O(\sqrt{\gamma})),$$

and thus $\|x\|^2 \approx \sum_{j=0}^{n-1} 2^j (\sqrt{8}/4)^{2j} (1 + O(\sqrt{\gamma}))^{2j} x_{i^\star}^2 =$

$nx_{i^\star}^2 \exp(O(n\sqrt{\gamma}))$. Choosing $\gamma = O(1/n^2)$ as suggested in Lemma 2.4 thus recovers the desirable guarantee $\|x\|^2 = O(x_{i^\star}^2 n)$ while yielding a condition number of no more than $O(n^2)$. Since the size of the matrix is $2^{2n} \times 2^{2n}$, this is only polylogarithmic in the matrix size. This completes the intuition behind our choice of parameters and results.

What remains is thus to formalize the above intuition and prove Lemma 2.3 and Lemma 2.4. This is postponed to Section B in the appendix. Similarly, in appendix Section A, we give the alternative reduction from $k$-Forrelation.

## 3. Conclusion

In this work, we give the first exponential separation between the influential quantum-inspired classical algorithms and quantum algorithms. We prove the separation for the fundamental problem of solving linear systems.

We give two alternative proofs, using a reduction either from a random walk problem in two binary trees or from the $k$-Forrelation problem. We find that both reductions have value, with the random walk reduction revealing interesting structural insights into a classic construction of two binary trees with randomly joined leaves. We hope these insights may prove useful in future work.

An interesting direction for future work is to prove similar separations for other natural problems. In particular, does there exists problems that are not BQP-complete, but where an exponential separation still exists? Another direction for future work is to identify properties of practically occurring linear systems that may be exploited to develop faster practical quantum-inspired classical algorithms. In light of our lower bounds, these properties inherently have to go beyond mere spectral properties of the linear system.

## Acknowledgment

Kasper Green Larsen is funded by the European Union (ERC, TUCLA, 101125203). Views and opinions expressed are however those of the author(s) only and do not necessarily reflect those of the European Union or the European Research Council. Neither the European Union nor the granting authority can be held responsible for them.

## Impact Statement

This paper presents work whose goal is to advance the field of Machine Learning. There are many potential societal consequences of our work, none which we feel must be specifically highlighted here.

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

## A. Reduction from $k$-Forrelation

Here we give an alternative reduction, starting from the $k$-Forrelation problem (Aaronson & Ambainis, 2015), which exhibits a maximum query complexity separation between quantum and randomized classical algorithms (Bansal & Sinha, 2021; Sherstov et al., 2021), and using the ideas in the proof by Harrow et al. (2009) that solving linear systems is BQP-complete. We start by recalling the main ideas in the two works.

**HHL's BQP-Hardness Reduction.** In (Harrow et al., 2009), the authors consider a general reduction for simulating any quantum circuit via solving a linear system. We will outline their approach here using their notation. Assume we have a quantum circuit acting on $n$ qubits. Let $U_1, \ldots, U_T$ denote the sequence of $2^n \times 2^n$ unitaries corresponding to the sequence of one- or two-qubit gates applied by the circuit on the initial state $|0\rangle^{\otimes n}$. They now consider the matrix

$$U = \sum_{t=1}^{T} |t\rangle\langle t-1| \otimes U_t + |t+T\rangle\langle t+T-1| \otimes I + |t+2T \bmod 3T\rangle\langle t+2T-1| \otimes U_{T+1-t}^{\dagger}. \tag{4}$$

For intuition on this construction, think of the matrix as being partitioned into a $3T \times 3T$ grid of sub-matrices of size $2^n \times 2^n$ each and notice how $|i\rangle\langle j| \otimes A$ places the matrix $A$ as the sub-matrix in position $(i, j)$ (with 0-indexed rows and columns $i, j \in [3T]$). A $3T \cdot 2^n$-dimensional vector is similarly partitioned into $3T$ blocks. For $1 \leq t \leq T$, observe that the vector $U^t|0\rangle|0\rangle^{\otimes n}$ has the state of the $n$ qubits after the first $t$ steps of computation stored in its $t+1$'st block, and all other blocks are 0. Note that the first $|0\rangle$ in $|0\rangle|0\rangle^{\otimes n}$ refers to the $3T$-dimensional standard unit vector with a 1 in its first entry. For $T \leq t \leq 2T-1$, $U^t|0\rangle|0\rangle^{\otimes n}$ simply has the final state of the qubits in the $t+1$'st block. Finally, for $2T \leq t \leq 3T-1$, the $t+1$'st block is undoing the computations until finally wrapping back around and recovering the initial state $|0\rangle|0\rangle^{\otimes n}$.

Harrow, Hassidim and Lloyd now consider the following matrix $A = I - Ue^{-1/T}$ and observe that its condition number is $O(T)$ and its inverse is given by

$$A^{-1} = \sum_{k=0}^{\infty} U^k e^{-k/T}. \tag{5}$$

Hence if we consider the solution to the linear system $Ax = |0\rangle|0\rangle^{\otimes n}$, then it is given by

$$x = A^{-1}|0\rangle|0\rangle^{\otimes n} = \sum_{k=0}^{\infty} U^k e^{-k/T}|0\rangle|0\rangle^{\otimes n} = \sum_{t=0}^{3T-1} U^t|0\rangle|0\rangle^{\otimes n} \sum_{k=0}^{\infty} e^{-(t+3Tk)/T}.$$

The crucial observation is now that the terms $U^t|0\rangle|0\rangle^{\otimes}$ for $t = T, \ldots, 2T-1$ has the final state of the computation in its $t+1$'st block and 0 elsewhere. Furthermore, since the $U^t$'s are unitaries and the $U^t|0\rangle|0\rangle^{\otimes}$ are disjoint in their non-zeros, it follows that if we sample and index $i$ from $x$ with probability $x_i^2/\|x\|^2$, then the sample is from one of the blocks $T+1, \ldots, 2T$ with probability

$$\frac{\sum_{t=T}^{2T-1} \left(\sum_{k=0}^{\infty} e^{-(t+3Tk)/T}\right)^2}{\sum_{t=1}^{3T-1} \left(\sum_{k=0}^{\infty} e^{-(t+3Tk)/T}\right)^2} = \Omega(1). \tag{6}$$

When this happens, the remaining $n$ bits in the index $i$ is a sample from the output state of the circuit. To finally obtain a symmetric matrix $M$, they define it as

$$M = \begin{pmatrix} 0 & A \\ A^{\dagger} & 0 \end{pmatrix} \tag{7}$$

and remark that its inverse is

$$M^{-1} = \begin{pmatrix} 0 & (A^{-1})^{\dagger} \\ A^{-1} & 0 \end{pmatrix}. \tag{8}$$

$k$-**Forrelation.** We now want to use the above reduction in combination with an oracle query problem to prove our lower bound for QIC algorithms. For this, consider the $k$-Forrelation problem. In this problem, the input is $k$ vectors $z^{(1)}, \ldots, z^{(k)} \in \{-1, 1\}^{2^n}$ and the goal is to estimate

$$\phi_{n,k}(z^{(1)}, \ldots, z^{(k)}) := \frac{1}{2^n} \mathbf{1}^T D_{z^{(1)}} H D_{z^{(2)}} \cdots H D_{z^{(k)}} \mathbf{1}.$$

Here $D_{z^{(i)}}$ is a diagonal matrix with $z^{(i)}$ on the diagonal, $\mathbf{1}$ is the all-1 vector and $H$ is the $2^n \times 2^n$ Hadamard matrix. The goal is to distinguish the two cases $|\phi_{n,k}(z^{(1)}, \ldots, z^{(k)})| \le \alpha$ and $|\phi_{n,k}(z^{(1)}, \ldots, z^{(k)})| \ge \beta$ for $0 < \alpha < \beta < 1$ chosen appropriately.

The definition of $\phi_{n,k}$ immediately gives a quantum circuit $\mathcal{C}_{k\text{-for}}$ for the problem using $k$ queries and $n$ qubits. On initial state $|0\rangle^{\otimes n}$, we can first use $n$ one-qubit Hadamard gates to transform the state to $\mathbf{1} \cdot 2^{-n/2}$. We next use the convention that an oracle query to an $z^{(i)}$ in the basis state $|a\rangle$, with $a$ an index in $[2^n]$, returns the state $z_a^{(i)}|a\rangle$. Each of the $D_{z^{(i)}}$'s are thus replaced by such oracle calls and each $H$ is implemented via $n$ one-qubit Hadamard gates. Finally, on the state $D_{z^{(1)}} H D_{z^{(2)}} \cdots H D_{z^{(k)}} \mathbf{1} \cdot 2^{-n/2}$, if we now apply another $2^n \times 2^n$ Hadamard matrix, then the value of the coordinate corresponding to $|0\rangle^{\otimes n}$ is precisely $\phi_{n,k}(z^{(1)}, \ldots, z^{(k)})$. Measuring the $n$ qubits thus returns the all-0 state with probability at least $\beta^2$ if $|\phi_{n,k}(z^{(1)}, \ldots, z^{(k)})| \ge \beta$ and at most $\alpha^2$ if $|\phi_{n,k}(z^{(1)}, \ldots, z^{(k)})| \le \alpha$.

For the particular choice of $\alpha = 2^{-5k-1}$ and $\beta = 2^{-5k}$, it thus suffices to run the quantum circuit $2^{O(k)}$ times to be able to distinguish the two cases except with a failure probability of $1/3$. However, for the same choice of $\alpha$ and $\beta$ (Bansal & Sinha, 2021; Sherstov et al., 2021), it was shown that any classical randomized algorithm with failure probability $1/3$ must make $\Omega((2^n/n)^{1-1/k})$ queries when $k$ is chosen as a constant.

**Combining HHL and $k$-Forrelation.** We now let $U_1, \ldots, U_T$ in the HHL reduction be the unitaries in the quantum circuit $\mathcal{C}_{k\text{-for}}$ for $k$-Forrelation, with each oracle query to a $z^{(i)}$ replaced by the diagonal matrix $D_{z^{(i)}}$. Thus each $U_j$ is either a one-qubit Hadamard gate or a diagonal matrix $D_{z^{(i)}}$. We thus have $T = (k+1)n + k$ (that is, $k+1$ Hadamard matrices using $n$ gates each and $k$ diagonal matrices) and may define the matrices $A$ and $M$ from (7). For constant $k$, the matrix $M$ is an $N \times N$ real symmetric matrix with $N = O(n2^n)$ and condition number $O(n) = O(\ln N)$.

Similarly to the reduction we gave in Section 2.1, assume we have a QIC algorithm $\mathcal{A}$ that makes $T(M, \varepsilon)$ queries to $SQ(M)$ to sample an index from $\tilde{x}$ such that each index $i$ is sampled with probability $\tilde{x}_i^2/\|\tilde{x}\|^2$ for a $\tilde{x}$ satisfying $\|\tilde{x} - x\| \le \varepsilon \|x\|$ for a sufficiently small $\varepsilon > 0$. Here $x = M^{-1}e_1$ is the solution to the linear system $Mx = e_1$. By the structure of $M^{-1}$ from (8) and $A^{-1}$ from (5), we see that

$$x = M^{-1}e_1 = \begin{pmatrix} 0 \\ A^{-1}|0\rangle|0\rangle^{\otimes n} \end{pmatrix}.$$

Here the first $|0\rangle$ in $A^{-1}|0\rangle|0\rangle^{\otimes n}$ is the $3T$-dimensional vector with a 1 in its first entry and 0 elsewhere.

We will now simulate the QIC algorithm by making classical queries to the inputs $z^{(1)}, \ldots, z^{(k)}$. For this, we first observe that every row and every column of $U$ (see (4)) has unit length and all its diagonal entries are 0. This further implies that the rows and columns of $A = I - Ue^{-1/T}$ and $M$ all have the same norm. Thus if the QIC algorithm $\mathcal{A}$ samples a row or column, we can simply return a uniform random index of a row or column. Next, if $\mathcal{A}$ samples an index in a row or column proportional to the square of the index, or queries an index, then we have two cases. If the row or column does not correspond to a diagonal $D_{z^{(i)}}$, then we can directly determine the entries of the row or column and simulate the sample or query without any oracle queries to $z^{(1)}, \ldots, z^{(k)}$. Finally, if the row or column corresponds to a row or column of $D_{z^{(i)}}$, we query the corresponding $z^{(i)}$ and thereby determine the diagonal entry of $D_{z^{(i)}}$. We can again correctly simulate a sample or query to the row/column. Thus the number of classical queries to $z^{(1)}, \ldots, z^{(k)}$ is at most $T(M, \varepsilon)$ to draw one sample from $\tilde{x}$.

Recall now that we can think of the vector $y = A^{-1}|0\rangle|0\rangle^{\otimes n}$ as consisting of $3T$ blocks $y^{(t)}$ with $y = \sum_{t=0}^{3T-1} |t\rangle \otimes y^{(t)}$ and each $y^{(t)}$ a $2^n$-dimensional vector (corresponding to the $n$ qubits). Furthermore, $y^{(T)}, \ldots, y^{(2T-1)}$ all equal the final state of the quantum $k$-Forrelation circuit $\mathcal{C}_{k\text{-for}}$. If we now use $\mathcal{A}$ to draw an index $i = |t\rangle|j\rangle$ from $\tilde{x}$ (with $t \in [3T], j \in [2^n]$), we first check if $t \in \{T, \ldots, 2T-1\}$. Assume for now that $\tilde{x} = x$. Then we know from (6) that $t \in \{T, \ldots, 2T-1\}$ with $\Omega(1)$ probability. If this is the case, we know that $j$ is the all-0 bit string with probability $\phi_{n,k}(z^{(1)}, \ldots, z^{(k)})^2$.

We now repeat the above sampling for $2^{O(k)}$ times with independent randomness and have obtained a classical algorithm distinguishing $|\phi_{n,k}(z^{(1)}, \ldots, z^{(k)})| \le 2^{-5k-1}$ from $|\phi_{n,k}(z^{(1)}, \ldots, z^{(k)})| \ge 2^{-5k}$ except with failure probability $1/3$ (via Hoeffding's inequality). The classical algorithm makes $2^{O(k)}T(M, \varepsilon)$ queries. Thus the lower bounds from previous works imply that $T(M, \varepsilon) = \Omega((2^n/n)^{1-1/k})$ for any constant choice of $k$.

**Handling Approximation.** In the above, we assumed samples were from the exact solution $x$. We now show how to handle samples from a $\tilde{x}$ with $\|x - \tilde{x}\| \le \varepsilon \|x\|$.

Recall from above that our obtained classical $k$-Forrelation algorithm performs $Q = 2^{O(k)}$ independent samples from the distribution corresponding to $x_i^2/\|x\|^2$. With a constant factor increase in samples, we may further assume the failure probability is at most $1/4$ instead of $1/3$. Now if the distribution given by $\tilde{x}_i^2/\|\tilde{x}\|^2$ has total variation distance $D$ from the distribution $x_i^2/\|x\|^2$, then the total variation distance on the full $Q$ samples is at most $QD$. If this is less than $1/12$, we get that the failure probability is at most $1/4 + 1/12 \le 1/3$ and the lower bound $T(M, \varepsilon) = \Omega((2^n/n)^{1-1/k})$ also applies to sampling from $\tilde{x}_i^2/\|\tilde{x}\|^2$ (assuming constant $k$).

Let us use $x^2$ to denote the vector with entries $x_i^2$ and $\tilde{x}^2$ the vector with entries $\tilde{x}_i^2$. Recalling that total variation distance is half the $\ell_1$ distance, we get

$$
\begin{aligned}
D &= \frac{1}{2} \cdot \left\| \frac{x^2}{\|x\|^2} - \frac{\tilde{x}^2}{\|\tilde{x}\|^2} \right\|_1 \\
&= \frac{1}{2} \cdot \left\| \frac{x^2}{\|x\|^2} - \frac{\tilde{x}^2}{\|x\|^2} \cdot \frac{\|\tilde{x}\|^2}{\|x\|^2} \right\|_1 \\
&= \frac{1}{2} \cdot \left\| \frac{x^2}{\|x\|^2} - \frac{\tilde{x}^2}{\|x\|^2} \cdot \left( 1 + \frac{\|\tilde{x}\|^2 - \|x\|^2}{\|x\|^2} \right) \right\|_1 \\
&\le \frac{1}{2} \cdot \left( \left\| \frac{x^2 - \tilde{x}^2}{\|x\|^2} \right\|_1 + \frac{|\|\tilde{x}\|^2 - \|x\|^2|}{\|x\|^2} \right).
\end{aligned}
$$

For $0 < \varepsilon \le 1$ we have that $\|\tilde{x}\|^2 = \|x + (\tilde{x} - x)\|^2 \le (\|x\| + \|\tilde{x} - x\|)^2 \le (\|x\| + \varepsilon\|x\|)^2 = (1+\varepsilon)^2\|x\|^2 \le (1+4\varepsilon)\|x\|^2$. Similarly, $\|\tilde{x}\|^2 = \|x + (\tilde{x} - x)\|^2 \ge (\|x\| - \|\tilde{x} - x\|)^2 \ge (1 - \varepsilon)^2\|x\|^2 \ge (1 - 2\varepsilon)\|x\|^2$. Hence

$$
D \le \frac{1}{2} \cdot \left( \frac{\|x^2 - \tilde{x}^2\|_1}{\|x\|^2} + 4\varepsilon \right).
$$

We also have $|x_i^2 - \tilde{x}_i^2| = |x_i^2 - (x_i + (\tilde{x}_i - x_i))^2| = |2x_i(\tilde{x}_i - x_i) - (\tilde{x}_i - x_i)^2| \le 2|x_i(\tilde{x}_i - x_i)| + (\tilde{x}_i - x_i)^2$. By Cauchy-Schwartz, we see that

$$
\sum_i |x_i(\tilde{x}_i - x_i)| \le \|x\| \cdot \|\tilde{x} - x\| \le \varepsilon\|x\|^2.
$$

We also have $\sum_i (\tilde{x}_i - x_i)^2 = \|\tilde{x} - x\|^2 \le \varepsilon^2\|x\|^2 \le \varepsilon\|x\|^2$. In conclusion

$$
D \le \frac{1}{2} \cdot \left( \frac{3\varepsilon\|x\|^2}{\|x\|^2} + 4\varepsilon \right)
$$

$$
\le 6\varepsilon.
$$

It follows that for $\varepsilon = 2^{-\Omega(k)}$, we have the lower bound $T(M, \varepsilon) = \Omega((2^n/n)^{1-1/k})$. Since $N = n2^n$, this is $T(M, \varepsilon) = \Omega((N/\ln^2 N)^{1-1/k})$. Rescaling $k$ by a factor 2, this simplifies to $T(M, \varepsilon) = \Omega(N^{1-1/k})$ when $\varepsilon = 2^{-\Omega(k)}$.

This gives us our main result in Theorem 1.5.

## B. Spectral Properties of Random Walk Matrix

In this section we Lemma 2.3 and Lemma 2.4 needed for our reduction from the random walk problem by Childs et al.

### B.1. Bounding the Eigenvalues of the Adjacency Matrix

We start by proving Lemma 2.3, i.e. that the adjacency matrix $A$ of the graph $G_n$ has no eigenvalues in the range $(\sqrt{8}, 3 - 2^{-n}]$.

Let us first introduce some terminology. We say that the root of $T_1$ is at level 1, the leaves of $T_1$ at level $n + 1$, the leaves of $T_2$ at level $n + 2$ and the root of $T_2$ is at level $2n + 2$. Note that this is unlike the definition of $\psi_j^{(i)}$ in the previous section, where we focused on a single tree $T_i$ and counted the level from the root of $T_i$ and down. From hereon, it is more convenient for us to think of the level as the distance from the root of $T_1$ (with the root at level 1). The two roots have degree two and all other nodes have degree three.

To bound the eigenvalues of $A$, we begin with an observation that allows us to restrict our attention to so-called *layered* eigenvectors (inspired by previous works bounding eigenvalues of trees (Hoory et al., 2006; Alon et al., 2021)). We say that an eigenvector $w$ is a layered eigenvector, if all entries of $w$ corresponding to nodes at level $i$ have the same value $\phi_i$. We argue that

**Observation B.1.** *If $A$ has an eigenvector of eigenvalue $\lambda$, then it has a real-valued layered eigenvector of eigenvalue $\lambda$.*

*Proof.* Let $u$ be an arbitrary eigenvector with eigenvalue $\lambda$. Since $A$ is real and symmetric, its eigenvalues are real. If $u$ is complex, we may write it as $u = v + ia$ with $v$ and $a$ real vectors. Since $Av + iAa = Au = \lambda u = \lambda v + i\lambda a$, it must be the case that $Av = \lambda v$ as these are the only real vectors in $Av + iAa$ and $\lambda v + i\lambda a$. By the same argument, we also have $Aa = \lambda a$. Thus $v$ and $a$ are real eigenvectors with eigenvalue $\lambda$. Assume wlog. that $v$ is non-zero.

We now show how to construct a real layered eigenvector $w$ from $v$ having the same eigenvalue $\lambda$. First recall that a vector $v$ is an eigenvector of $A$ with eigenvalue $\lambda$, if and only if $(\lambda I - A)v = 0$. Now given the eigenvector $v$, let $S_i$ be the set of coordinates corresponding to nodes in level $i$ and let $\phi_i = w_j = |S_i|^{-1} \sum_{k \in S_i} v_k$ for all $j \in S_i$. We will verify that $(\lambda I - A)w = 0$, which implies that $w$ is also an eigenvector of eigenvalue $\lambda$.

For this, consider a level $i$ and define $p_i \in \{0, 1, 2\}$ as the number of neighbors a node in level $i$ has in level $i-1$. Similarly, define $c_i \in \{0, 1, 2\}$ as the number of neighbors a node in level $i$ has in level $i+1$. To verify that $(\lambda I - A)w = 0$, we verify that for all levels $i$, we have $\lambda \phi_i - p_i \phi_{i-1} - c_i \phi_{i+1} = 0$. To see this, observe that since $(\lambda I - A)v = 0$, it holds for every index $j \in S_i$ that $\lambda v_j - \sum_{k \in S_{i-1}: j \in \mathcal{N}(k)} v_k - \sum_{k \in S_{i+1}: j \in \mathcal{N}(k)} v_k = 0$, where $\mathcal{N}(k)$ denotes the neighbors of $k$. Summing this across all nodes $j \in S_i$, we see that $\lambda \sum_{k \in S_i} v_k - c_{i-1} \sum_{k \in S_{i-1}} v_k - p_{i+1} \sum_{k \in S_{i+1}} v_k = 0$. This follows since every node $k$ in level $i-1$ has $v_k$ included precisely $c_{i-1}$ times in the sum ($p_{i+1}$ times for nodes in level $i + 1$).

The left hand side also equals $\lambda \phi_i |S_i| - c_{i-1} \phi_{i-1} |S_{i-1}| - p_{i+1} \phi_{i+1} |S_{i+1}|$ by definition of $\phi_i$. Now observe that $c_{i-1} |S_{i-1}| = p_i |S_i|$ since both sides of the equality equals the number of edges between levels $i-1$ and $i$. Using this, we rewrite $0 = \lambda \phi_i |S_i| - c_{i-1} \phi_{i-1} |S_{i-1}| - p_{i+1} \phi_{i+1} |S_{i+1}| = \lambda \phi_i |S_i| - p_i \phi_{i-1} |S_i| - c_i \phi_{i+1} |S_i| = |S_i|(\lambda \phi_i - p_i \phi_{i-1} - c_i \phi_{i+1})|S_i|$. Since $|S_i| \neq 0$, we conclude $\lambda \phi_i - p_i \phi_{i-1} - c_i \phi_{i+1} = 0$. $\square$

Using Observation B.1 it thus suffices for us to argue that $A$ has no real-valued layered eigenvectors with eigenvalues in the range $(\sqrt{8}, 3 - 2^{-n}]$. The first step in this argument, is to lower bound the largest eigenvalue $\lambda_1$. For this, recall that $\lambda_1 = \max_{v:\|v\|=1} v^T A v$. Noting that $A$ is $N \times N$ with $N = 2^{n+2} - 2$, we now consider the unit vector $v$ with all entries $1/\sqrt{N}$. Then $(Av)_i = 3/\sqrt{N}$ for $i$ not one of the two roots, and $(Av)_i = 2/\sqrt{N}$ for $i$ one of the two roots. Hence

$$\lambda_1 \geq 3 - 2/N > 3 - 2^{-n}. \tag{9}$$

Next, we aim to understand the structure of all real-valued layered eigenvectors of eigenvalues larger than $\sqrt{8}$. So let $w$ be an arbitrary such eigenvector of eigenvalue $\lambda > \sqrt{8}$ and let $\phi_i$ denote the value of the coordinates of $w$ corresponding to level $i$. Our goal is to show that $\lambda > \sqrt{8}$ implies $\lambda > 3 - 2^{-n}$.

Since $(\lambda I - A)w = 0$, we conclude from the row of $A$ corresponding to the root of $T_1$ that $\lambda \phi_1 - 2\phi_2 = 0 \Rightarrow \phi_2 = (\lambda/2)\phi_1$. For levels $1 \leq i \leq n$, we must have $\lambda \phi_{i+1} - \phi_i - 2\phi_{i+2} = 0$. As discussed earlier, a recurrence $a\phi_{i+2} + b\phi_{i+1} + c\phi_i = 0$ is known as a second degree linear recurrence and when $b^2 > 4ac$ its (real) solutions are of the form

$$\phi_i = \alpha \cdot \left(\frac{-b + \sqrt{b^2 - 4ac}}{2a}\right)^i + \beta \cdot \left(\frac{-b - \sqrt{b^2 - 4ac}}{2a}\right)^i,$$

with $\alpha, \beta \in \mathbb{R}$ such that the initial condition $\phi_2 = (\lambda/2)\phi_1$ is satisfied. Note that in our case, $a = -2, b = \lambda$ and $c = -1$. Since we assume $\lambda > \sqrt{8}$, we indeed have $b^2 > 4ac$. It follows that

$$\phi_i = \alpha \cdot \left(\frac{\lambda - \sqrt{\lambda^2 - 8}}{4}\right)^i + \beta \cdot \left(\frac{\lambda + \sqrt{\lambda^2 - 8}}{4}\right)^i,$$

for $\alpha, \beta$ satisfying $\phi_2 = (\lambda/2)\phi_1$. For short, let us define $\Delta = \sqrt{\lambda^2 - 8}$. Note that $\Delta$ is real since we assume $\lambda > \sqrt{8}$. Furthermore, we have $\lambda - \Delta > 0$ and thus in later equations, it is safe to divide by $\lambda - \Delta$.

The initial condition now implies that we must have

$$
\begin{aligned}
\alpha \cdot \left(\frac{\lambda - \Delta}{4}\right)^2 + \beta \cdot \left(\frac{\lambda + \Delta}{4}\right)^2 &= (\lambda/2)\left(\alpha \cdot \frac{\lambda - \Delta}{4} + \beta \cdot \frac{\lambda + \Delta}{4}\right) \Rightarrow \\
\alpha \cdot (\lambda - \Delta)^2 + \beta \cdot (\lambda + \Delta)^2 &= 2\lambda\left(\alpha \cdot (\lambda - \Delta) + \beta \cdot (\lambda + \Delta)\right) \Rightarrow \\
\beta \cdot \left(\lambda^2 + \Delta^2 + 2\lambda\Delta - 2\lambda(\lambda + \Delta)\right) &= \alpha \cdot \left(-\lambda^2 - \Delta^2 + 2\lambda\Delta + 2\lambda(\lambda - \Delta)\right) \Rightarrow \\
\beta \cdot \left(-\lambda^2 + \Delta^2\right) &= \alpha \cdot \left(\lambda^2 - \Delta^2\right) \Rightarrow \\
\beta &= -\alpha.
\end{aligned}
$$

Thus the solutions have the following form for any $\alpha \in \mathbb{R}$:

$$
\begin{aligned}
\phi_i &= \alpha \cdot \left(\left(\frac{\lambda - \Delta}{4}\right)^i - \left(\frac{\lambda + \Delta}{4}\right)^i\right) \\
&= \alpha \left(\frac{\lambda - \Delta}{4}\right)^i \cdot \left(1 - \left(\frac{\lambda + \Delta}{\lambda - \Delta}\right)^i\right)
\end{aligned}
\tag{10}
$$

for $1 \le i \le n + 2$.

Now observe that the same equations apply starting from the root of $T_2$ and down towards the leaves of $T_2$ (recall our convention that the root of $T_2$ is at level $2n + 2$), thus for $1 \le i \le n + 2$

$$
\phi_{2n+3-i} = \alpha' \left(\frac{\lambda - \Delta}{4}\right)^i \cdot \left(1 - \left(\frac{\lambda + \Delta}{\lambda - \Delta}\right)^i\right).
\tag{11}
$$

Examining (10) and (11), we first notice that

$$
\left(\frac{\lambda - \Delta}{4}\right)^i \cdot \left(1 - \left(\frac{\lambda + \Delta}{\lambda - \Delta}\right)^i\right) < 0.
\tag{12}
$$

Furthermore, both (10) and (11) give bounds on the leaf levels $\phi_{n+1}$ and $\phi_{n+2}$ and these must be equal. Any non-zero solution thus must have $\alpha \cdot \alpha' > 0$, i.e. the two have the same sign. From (10), (11) and (12) it now follows that all pairs $\phi_i$ and $\phi_j$ satisfy $\phi_i \cdot \phi_j > 0$.

Now let $v_1$ be a layered and real-valued eigenvector corresponding to the largest eigenvalue $\lambda_1$ (such a layered eigenvector is guaranteed to exist by Observation B.1). Let $\psi_i$ denote the values of the entries in $v_1$ corresponding to nodes of level $i$. Since $\lambda_1 > \sqrt{8}$, the values $\psi_i$ satisfy $\psi_i \cdot \psi_j > 0$ for all pairs $\psi_i$ and $\psi_j$ by the above arguments. Since eigenvectors corresponding to distinct eigenvalues of real symmetric matrices are orthogonal, we must have that either $\lambda = \lambda_1$ or $v_1^T w = 0$. But

$$
\begin{aligned}
(v_1^T w)^2 &= \left(\sum_{i=1}^{2n+2} 2^{\min\{i-1, 2n+2-i\}} \phi_i \psi_i\right)^2 \\
&= \sum_{i=1}^{2n+2}\sum_{j=1}^{2n+2} 2^{\min\{i-1, 2n+2-i\}} 2^{\min\{j-1, 2n+2-j\}} \phi_i \phi_j \psi_i \psi_j \\
&> 0.
\end{aligned}
$$

We thus conclude $\lambda = \lambda_1 > 3 - 2^{-n}$.

## B.2. Understanding the Solution

In this section, we prove Lemma 2.4. That is, we show that for $\lambda = \sqrt{8} + \gamma$ with $0 < \gamma \le \left(\frac{1}{16(n+2)}\right)^2$, it holds that $x_{i^\star}^2 = \Omega(n^{-5}\|x\|^2)$ where $x = M^{-1}e_1$ is the solution to the linear system $Mx = e_1$ and $i^\star$ is the index of the root of $T_2$.

As remarked earlier, we have that the matrix $M$ is block diagonal, where one block corresponds to the matrix $B = \lambda I - A$, where $A$ is the adjacency matrix of $G_n$. Furthermore, $e_1$ is non-zero only in the block corresponding to $B$. It follows that it suffices to understand the solution $x = B^{-1}e_1$ as padding it with zeroes gives $M^{-1}e_1$.

Similarly to the previous section, we will focus on *layered* vectors $x$. More formally, we claim there is a layered vector $x$ such that $x = B^{-1}e_1$. That is, all entries of $x$ corresponding to nodes in level $i$ have the same value $\phi_i$. Since $B$ is full rank, the solution $x$ to $Bx = e_1$ is unique and thus must equal this layered vector. We thus set out to determine appropriate values $\phi_i$.

Similarly to the previous section, we have that choosing $\phi_i$'s of the form

$$\phi_i = \alpha \left(\frac{\lambda - \Delta}{4}\right)^i + \beta \left(\frac{\lambda + \Delta}{4}\right)^i,$$

for any $\alpha, \beta \in \mathbb{R}$ and $i = 1, \ldots, n+2$ satisfy the recurrence $-2\phi_{i+2} + \lambda\phi_{i+1} - \phi_i = 0$. However, the initial condition we get from $Bx = e_1$ (the first row/equality of the linear system) is different than the previous section. Concretely, the initial condition becomes $\lambda\phi_1 - 2\phi_2 = 1 \Rightarrow \phi_2 = (\lambda/2)\phi_1 - 1/2$. This forces $\beta$ to be chosen as

$$
\begin{aligned}
\alpha \cdot \left(\frac{\lambda - \Delta}{4}\right)^2 + \beta \cdot \left(\frac{\lambda + \Delta}{4}\right)^2 &= (\lambda/2)\left(\alpha \cdot \frac{\lambda - \Delta}{4} + \beta \cdot \frac{\lambda + \Delta}{4}\right) - 1/2 \Rightarrow \\
\alpha \cdot (\lambda - \Delta)^2 + \beta \cdot (\lambda + \Delta)^2 &= (2\lambda)\left(\alpha(\lambda - \Delta) + \beta(\lambda + \Delta)\right) - 8 \Rightarrow \\
\beta \cdot \left(\lambda^2 + \Delta^2 + 2\lambda\Delta - 2\lambda(\lambda + \Delta)\right) &= \alpha \cdot \left(-\lambda^2 - \Delta^2 + 2\lambda\Delta + 2\lambda(\lambda - \Delta)\right) - 8 \Rightarrow \\
\beta \cdot \left(-\lambda^2 + \Delta^2\right) &= \alpha \cdot \left(\lambda^2 - \Delta^2\right) - 8 \Rightarrow \\
-8\beta &= 8\alpha - 8 \Rightarrow \\
\beta &= -\alpha + 1.
\end{aligned}
$$

Thus $\phi_i$'s of the following form, for $1 \le i \le n+2$ and $\alpha \in R$, satisfy the constraints corresponding to $T_1$

$$
\begin{aligned}
\phi_i &= \alpha \cdot \left(\left(\frac{\lambda - \Delta}{4}\right)^i - \left(\frac{\lambda + \Delta}{4}\right)^i\right) + \left(\frac{\lambda + \Delta}{4}\right)^i \\
&= \alpha \cdot \left(\frac{\lambda - \Delta}{4}\right)^i \left(1 - \left(\frac{\lambda + \Delta}{\lambda - \Delta}\right)^i\right) + \left(\frac{\lambda + \Delta}{4}\right)^i.
\end{aligned}
\tag{13}
$$

The tree $T_2$ puts similar constraints on $\phi_{2n+3-i}$ for $i = 1, \ldots, n+2$, except the root of $T_2$ gives the initial condition $\lambda\phi_{2n+2} - 2\phi_{2n+1} = 0$ (recall again that we say the root of $T_2$ is at level $2n+2$). This is precisely the same recurrence as in the previous section. Thus for $1 \le i \le n+2$ we have from (11) that for any $\alpha' \in \mathbb{R}$, the following satisfy the constraints of $T_2$

$$\phi_{2n+3-i} = \alpha' \cdot \left(\frac{\lambda - \Delta}{4}\right)^i \left(1 - \left(\frac{\lambda + \Delta}{\lambda - \Delta}\right)^i\right). \tag{14}$$

Noting that both (13) and (14) gives a formula for $\phi_{n+1}$ and $\phi_{n+2}$, we get two linear equations in two unknowns, i.e.

$$\alpha \cdot \left(\frac{\lambda - \Delta}{4}\right)^{n+j} \left(1 - \left(\frac{\lambda + \Delta}{\lambda - \Delta}\right)^{n+j}\right) + \left(\frac{\lambda + \Delta}{4}\right)^{n+j} = \alpha' \cdot \left(\frac{\lambda - \Delta}{4}\right)^{n+3-j} \left(1 - \left(\frac{\lambda + \Delta}{\lambda - \Delta}\right)^{n+3-j}\right).$$

for $j = 1, 2$. The equations have a unique solution $\alpha, \alpha'$ provided that the two vectors $v_j$

$$v_j = \left(\left(\frac{\lambda - \Delta}{4}\right)^{n+j} \left(1 - \left(\frac{\lambda + \Delta}{\lambda - \Delta}\right)^{n+j}\right), \left(\frac{\lambda - \Delta}{4}\right)^{n+3-j} \left(1 - \left(\frac{\lambda + \Delta}{\lambda - \Delta}\right)^{n+3-j}\right)\right)$$

with $j = 1, 2$ are linearly independent. To see that they are linearly independent, we compare the ratio $r_1 = v_1(1)/v_2(1)$ of the first coordinate $v_1$ and $v_2$ to the ratio $r_2 = v_1(2)/v_2(2)$ of the second coordinate. We have that $v_1$ and $v_2$ are linearly

independent if these ratios are distinct. The ratios of the coordinates are

$$
r_1 = \left( \frac{\lambda - \Delta}{4} \right)^{-1} \frac{\left( 1 - \left( \frac{\lambda+\Delta}{\lambda-\Delta} \right)^{n+1} \right)}{\left( 1 - \left( \frac{\lambda+\Delta}{\lambda-\Delta} \right)^{n+2} \right)}, r_2 = \left( \frac{\lambda - \Delta}{4} \right) \frac{\left( 1 - \left( \frac{\lambda+\Delta}{\lambda-\Delta} \right)^{n+2} \right)}{\left( 1 - \left( \frac{\lambda+\Delta}{\lambda-\Delta} \right)^{n+1} \right)}.
$$

We thus have $r_1 = r_2^{-1}$, implying linear independence whenever $|r_2| \neq 1$. To argue that this is the case, we prove the following auxiliary results using simple approximations of $(1 + x)^a$

**Claim B.2.** *For $i \leq n + 2 \leq \gamma^{-1/2}/16$ and $0 < \gamma \leq 1/64$, we have $\Delta \leq 4\sqrt{\gamma}$,*

1.

$$
0 < \frac{2\Delta}{\lambda + \Delta} < \frac{2\Delta}{\lambda - \Delta} < \frac{1}{2(n + 2)},
$$

2.

$$
1 + \frac{i\Delta}{\lambda - \Delta} \leq \left( \frac{\lambda + \Delta}{\lambda - \Delta} \right)^i \leq 1 + \frac{4i\Delta}{\lambda - \Delta}.
$$

3.

$$
1 - \frac{4i\Delta}{\lambda + \Delta} \leq \left( \frac{\lambda - \Delta}{\lambda + \Delta} \right)^i \leq 1 - \frac{i\Delta}{\lambda + \Delta}.
$$

4.

$$
1 + \frac{1}{2(n + 1)} \leq \frac{1 - \left( \frac{\lambda+\Delta}{\lambda-\Delta} \right)^{n+2}}{1 - \left( \frac{\lambda+\Delta}{\lambda-\Delta} \right)^{n+1}} \leq 1 + \frac{4}{n + 1}.
$$

From Claim B.2 point 4., $\lambda - \Delta \leq \lambda = \sqrt{8} + \gamma$, $n + 2 \leq \gamma^{-1/2}/16$, $\gamma \leq 1/64$ and $n$ large enough, we get

$$
0 < r_2 \leq \frac{\sqrt{8} + 1/64}{4} \cdot \left( 1 + \frac{4}{n + 1} \right) < 1,
$$

implying linear independence and thus existence of a unique solution $\alpha, \alpha'$. We defer the proof of Claim B.2 to Section B.3 as it consists of careful, but simple, calculations.

We now derive a number of properties of this unique choice of $\alpha$ and $\alpha'$. For this, we need one last auxiliary result

**Claim B.3.** *For $n + 2 \leq \gamma^{-1/2}/16$ and $0 < \gamma \leq 1/64$, we have*

$$
-1 - \frac{2}{n + 1} \leq \alpha \left( \left( \frac{\lambda - \Delta}{\lambda + \Delta} \right)^{n+1} - 1 \right) \leq -1 - \frac{1}{4(n + 1)}.
$$

We again defer the proof of Claim B.3 to Section B.3.

Our goal is now to bound all terms $\phi_i$ in terms of $\phi_{2n+2}$ as $\phi_{2n+2}$ is the value of the entry of $x$ corresponding to the root of $T_2$. We do this in two steps. First, we relate all $\phi_i$ to $\phi_{n+1}$ as both (13) and (14) gives a formula for $\phi_{n+1}$. Using the relationship between $\phi_{2n+2}$ and $\phi_{n+1}$ then indirectly relates all $\phi_i$ to $\phi_{2n+2}$.

For $i \leq n + 1$, we relate $\phi_{2n+3-i}$ to $\phi_{n+1}$ directly from (14), giving

$$
\frac{\phi_{2n+3-i}}{\phi_{n+1}} = \left( \frac{\lambda - \Delta}{4} \right)^{i-n-2} \cdot \frac{\left( \frac{\lambda+\Delta}{\lambda-\Delta} \right)^i - 1}{\left( \frac{\lambda+\Delta}{\lambda-\Delta} \right)^{n+2} - 1}.
$$

From Claim B.2, this implies

$$\left(\frac{\lambda - \Delta}{4}\right)^{i-n-2} \frac{i}{4(n+2)} \leq \frac{|\phi_{2n+3-i}|}{|\phi_{n+1}|} \leq \left(\frac{\lambda - \Delta}{4}\right)^{i-n-2} \frac{4i}{n+2}. \tag{15}$$

To bound the terms $\phi_i$ with $i \leq n + 1$, we get from (13) that

$$\frac{\phi_i}{\phi_{n+1}} = \left(\frac{\lambda + \Delta}{4}\right)^{i-n-1} \cdot \frac{\alpha\left(\left(\frac{\lambda-\Delta}{\lambda+\Delta}\right)^i - 1\right) + 1}{\alpha\left(\left(\frac{\lambda-\Delta}{\lambda+\Delta}\right)^{n+1} - 1\right) + 1}. \tag{16}$$

Using Claim B.3, we conclude

$$\begin{aligned}
\frac{|\phi_i|}{|\phi_{n+1}|} &\leq \left(\frac{\lambda + \Delta}{4}\right)^{i-n-1} \cdot \frac{\left|\alpha\left(\left(\frac{\lambda-\Delta}{\lambda+\Delta}\right)^i - 1\right)\right| + 1}{(4(n+1))^{-1}} \\
&\leq 4(n+1) \cdot \left(\frac{\lambda + \Delta}{4}\right)^{i-n-1} \cdot \left(\left|\alpha\left(\left(\frac{\lambda-\Delta}{\lambda+\Delta}\right)^{n+1} - 1\right)\right| + 1\right) \\
&\leq 12(n+1) \cdot \left(\frac{\lambda + \Delta}{4}\right)^{i-n-1}.
\end{aligned} \tag{17}$$

Note here that it was crucial that our bound in Claim B.3 was very tight since the denominator in (16) is very close to zero. From (15) with $i = 1$, we conclude

$$|\phi_{n+1}| \leq |\phi_{2n+2}|4(n+2)\left(\frac{\lambda - \Delta}{4}\right)^{n+1}. \tag{18}$$

Inserting (18) into (15), we now get for $i \leq n + 2$ that:

$$|\phi_{2n+3-i}| \leq 4(n+2)|\phi_{2n+2}|\left(\frac{\lambda - \Delta}{4}\right)^{i-1}\frac{4i}{n+2} = 16i|\phi_{2n+2}|\left(\frac{\lambda - \Delta}{4}\right)^{i-1}.$$

Similarly, inserting (18) into (17), we get

$$|\phi_i| \leq 4(n+2)|\phi_{2n+2}|12(n+1)\left(\frac{\lambda + \Delta}{4}\right)^i \leq 48(n+2)^2|\phi_{2n+2}|\left(\frac{\lambda + \Delta}{4}\right)^i.$$

Using that there are $2^{i-1}$ nodes at level $i$ and $2n + 3 - i$, we finally get that

$$
\begin{aligned}
\|x\|^2 &= \sum_{i=1}^{n+1} 2^{i-1} \left( \phi_i^2 + \phi_{2n+3-i}^2 \right) \\
&\leq \sum_{i=1}^{n+1} 2^{i-1} \left( (16i)^2 \left( \frac{\lambda - \Delta}{4} \right)^{2i-2} + 48^2 (n+2)^4 \left( \frac{\lambda + \Delta}{4} \right)^{2i} \right) \phi_{2n+2}^2 \\
&= O\left( n^4 \phi_{2n+2}^2 \sum_{i=1}^{n+1} 2^i \left( \frac{\lambda + \Delta}{4} \right)^{2i} \right) \\
&= O\left( n^4 \phi_{2n+2}^2 \sum_{i=1}^{n+1} 2^i \left( \frac{\lambda^2 + \Delta^2 + 2\lambda\Delta}{16} \right)^i \right) \\
&= O\left( n^4 \phi_{2n+2}^2 \sum_{i=1}^{n+1} 2^i \left( \frac{2(8 + 2\sqrt{8}\gamma + \gamma^2) - 8 + 2(\sqrt{8} + \gamma)4\sqrt{\gamma}}{16} \right)^i \right) \\
&= O\left( n^4 \phi_{2n+2}^2 \sum_{i=1}^{n+1} 2^i \left( \frac{8 + 48\sqrt{\gamma}}{16} \right)^i \right) \\
&= O\left( n^4 \phi_{2n+2}^2 \sum_{i=1}^{n+1} 2^i \left( \frac{1}{2}(1 + 6\sqrt{\gamma}) \right)^i \right) \\
&= O\left( n^5 \phi_{2n+2}^2 (1 + 6\sqrt{\gamma})^n \right) \\
&= O\left( n^5 \phi_{2n+2}^2 e^{6\sqrt{\gamma}n} \right).
\end{aligned}
$$

For $n + 2 \leq \gamma^{-1/2}/16$, this is $O(n^5 \phi_{2n+2}^2)$, implying $\phi_{2n+2}^2 = \Omega(n^{-5}\|x\|^2)$. Since $\phi_{2n+2}$ is the value $x_{i^\star}$ of the root $i^\star$ of $T_2$, this completes the proof.

### B.3. Auxiliary Results

In this section, we prove the two auxiliary claims in Claim B.2 and Claim B.3. The first claim uses simple approximations of $(1 + x)^a$ without any particularly novel ideas.

*Proof of Claim B.2.* Using a Taylor series, we have for $0 \leq x < 1$ that

$$
\begin{aligned}
1 + x &= \exp\left( \sum_{n=1}^{\infty} (-1)^{n+1} x^n / n \right) \\
&\geq \exp(x - x^2/2) \\
&\geq \exp(x/2).
\end{aligned}
$$

At the same time, it holds for all $x$ that $1 + x \leq e^x$. Now observe that $\Delta = \sqrt{(\sqrt{8} + \gamma)^2 - 8} = \sqrt{2\sqrt{8}\gamma + \gamma^2}$. Thus $\sqrt{\gamma} \leq \Delta \leq 4\sqrt{\gamma}$ when $\gamma \leq 1$. This implies for $\gamma \leq 1/64$ that

$$
\frac{2\Delta}{\lambda - \Delta} \leq \frac{8\sqrt{\gamma}}{\sqrt{8} - 4\sqrt{\gamma}} \leq \frac{8\sqrt{\gamma}}{\sqrt{8} - 1/2} \leq 8\sqrt{\gamma}. \tag{19}
$$

For integer $i \leq \gamma^{-1/2}/8$, we therefore have that $\left( \frac{\lambda + \Delta}{\lambda - \Delta} \right)^i = \left( 1 + \frac{2\Delta}{\lambda - \Delta} \right)^i$ satisfies

$$
1 + \frac{i\Delta}{\lambda - \Delta} \leq \exp\left( i \cdot \frac{\Delta}{\lambda - \Delta} \right) \leq \left( 1 + \frac{2\Delta}{\lambda - \Delta} \right)^i \leq \exp\left( i \cdot \frac{2\Delta}{\lambda - \Delta} \right) \leq 1 + \frac{4i\Delta}{\lambda - \Delta}. \tag{20}
$$

For $0 \leq x < 1/2$, we also have $1 - x \leq e^{-x}$ and $1 - x = \exp(-\sum_{n=1}^{\infty} x^n/n) \geq \exp(-\sum_{n=1}^{\infty} x(1/2)^{n-1}) \geq \exp(-2x)$. Since $0 < 2\Delta/(\lambda+\Delta) \leq 2\Delta/(\lambda-\Delta) \leq 8\sqrt{\gamma}$, we thus conclude for $i \leq \gamma^{-1/2}/16$ that $\left(\frac{\lambda-\Delta}{\lambda+\Delta}\right)^i = \left(1 - \frac{2\Delta}{\lambda+\Delta}\right)^i$ satisfies

$$1 - \frac{4i\Delta}{\lambda+\Delta} \leq \exp\left(-4i \cdot \frac{\Delta}{\lambda+\Delta}\right) \leq \left(1 - \frac{2\Delta}{\lambda+\Delta}\right)^i \leq \exp\left(-i \cdot \frac{2\Delta}{\lambda+\Delta}\right) \leq 1 - \frac{i\Delta}{\lambda+\Delta}.$$

We finally bound the ratio

$$\frac{1 - \left(\frac{\lambda+\Delta}{\lambda-\Delta}\right)^{n+2}}{1 - \left(\frac{\lambda+\Delta}{\lambda-\Delta}\right)^{n+1}} = \frac{1 - \left(\frac{\lambda+\Delta}{\lambda-\Delta}\right)^{n+1} - \frac{2\Delta}{\lambda-\Delta}\left(\frac{\lambda+\Delta}{\lambda-\Delta}\right)^{n+1}}{1 - \left(\frac{\lambda+\Delta}{\lambda-\Delta}\right)^{n+1}} = 1 + \frac{\frac{2\Delta}{\lambda-\Delta}\left(\frac{\lambda+\Delta}{\lambda-\Delta}\right)^{n+1}}{\left(\frac{\lambda+\Delta}{\lambda-\Delta}\right)^{n+1} - 1}.$$

From (20) and $n + 2 \leq \gamma^{-1/2}/16$, we have

$$\frac{\frac{2\Delta}{\lambda-\Delta}\left(\frac{\lambda+\Delta}{\lambda-\Delta}\right)^{n+1}}{\left(\frac{\lambda+\Delta}{\lambda-\Delta}\right)^{n+1} - 1} \geq \frac{\frac{2\Delta}{\lambda-\Delta}}{\left(\frac{\lambda+\Delta}{\lambda-\Delta}\right)^{n+1} - 1} \geq \frac{\frac{2\Delta}{\lambda-\Delta}}{\frac{4(n+1)\Delta}{\lambda-\Delta}} = \frac{1}{2(n+1)}.$$

From (20), (19) and $n + 2 \leq \gamma^{-1/2}/16$ we similarly get

$$\frac{\frac{2\Delta}{\lambda-\Delta}\left(\frac{\lambda+\Delta}{\lambda-\Delta}\right)^{n+1}}{\left(\frac{\lambda+\Delta}{\lambda-\Delta}\right)^{n+1} - 1} \leq \frac{\frac{2\Delta}{\lambda-\Delta}\left(1 + \frac{4(n+1)}{\lambda-\Delta}\right)}{\frac{(n+1)\Delta}{\lambda-\Delta}} \leq \frac{2}{n+1} + \frac{8\Delta}{\lambda-\Delta} \leq \frac{2}{n+1} + 32\sqrt{\gamma} \leq \frac{4}{n+1}.$$

Thus

$$1 + \frac{1}{2(n+1)} \leq \frac{1 - \left(\frac{\lambda+\Delta}{\lambda-\Delta}\right)^{n+2}}{1 - \left(\frac{\lambda+\Delta}{\lambda-\Delta}\right)^{n+1}} \leq 1 + \frac{4}{n+1}.$$

$\square$

We next proceed to give the proof of Claim B.3. They key idea here is to consider the ratio $\phi_{n+1}/\phi_{n+2}$ using both (13) and (14).

*Proof of Claim B.3.* Consider the ratio $\phi_{n+1}/\phi_{n+2}$. Using (13), this ratio equals

$$\frac{\phi_{n+1}}{\phi_{n+2}} = \frac{\left(\frac{\lambda-\Delta}{4}\right)^{n+1}\left(\alpha\left(1 - \left(\frac{\lambda+\Delta}{\lambda-\Delta}\right)^{n+1}\right) + \left(\frac{\lambda+\Delta}{\lambda-\Delta}\right)^{n+1}\right)}{\left(\frac{\lambda-\Delta}{4}\right)^{n+2}\left(\alpha\left(1 - \left(\frac{\lambda+\Delta}{\lambda-\Delta}\right)^{n+2}\right) + \left(\frac{\lambda+\Delta}{\lambda-\Delta}\right)^{n+2}\right)}.$$

This implies

$$\frac{\phi_{n+1}}{\phi_{n+2}} \cdot \left(\frac{\lambda-\Delta}{4}\right)\left(\alpha\left(1 - \left(\frac{\lambda+\Delta}{\lambda-\Delta}\right)^{n+2}\right) + \left(\frac{\lambda+\Delta}{\lambda-\Delta}\right)^{n+2}\right) = \left(\alpha\left(1 - \left(\frac{\lambda+\Delta}{\lambda-\Delta}\right)^{n+1}\right) + \left(\frac{\lambda+\Delta}{\lambda-\Delta}\right)^{n+1}\right).$$

Using (14), we also have

$$\frac{\phi_{n+1}}{\phi_{n+2}} = \frac{\lambda-\Delta}{4} \cdot \frac{1 - \left(\frac{\lambda+\Delta}{\lambda-\Delta}\right)^{n+2}}{1 - \left(\frac{\lambda+\Delta}{\lambda-\Delta}\right)^{n+1}}$$

Inserting this above and rearranging terms gives

$$\alpha\left(1-\left(\frac{\lambda+\Delta}{\lambda-\Delta}\right)^{n+1}\right)\left(\left(\frac{\lambda-\Delta}{4}\right)^2\left(\frac{1-\left(\frac{\lambda+\Delta}{\lambda-\Delta}\right)^{n+2}}{1-\left(\frac{\lambda+\Delta}{\lambda-\Delta}\right)^{n+1}}\right)^2-1\right)=$$
$$\left(\frac{\lambda+\Delta}{\lambda-\Delta}\right)^{n+1}\left(1-\left(\frac{\lambda+\Delta}{4}\right)\left(\frac{\lambda-\Delta}{4}\right)\frac{1-\left(\frac{\lambda+\Delta}{\lambda-\Delta}\right)^{n+2}}{1-\left(\frac{\lambda+\Delta}{\lambda-\Delta}\right)^{n+1}}\right). \tag{21}$$

Rearranging terms and using $\lambda^2-\Delta^2=8$ gives

$$\alpha\left(\left(\frac{\lambda-\Delta}{\lambda+\Delta}\right)^{n+1}-1\right)=\frac{1-\frac{1}{2}\cdot\frac{1-\left(\frac{\lambda+\Delta}{\lambda-\Delta}\right)^{n+2}}{1-\left(\frac{\lambda+\Delta}{\lambda-\Delta}\right)^{n+1}}}{\left(\frac{\lambda-\Delta}{4}\right)^2\left(\frac{1-\left(\frac{\lambda+\Delta}{\lambda-\Delta}\right)^{n+2}}{1-\left(\frac{\lambda+\Delta}{\lambda-\Delta}\right)^{n+1}}\right)^2-1}$$
$$=\frac{1}{2}\cdot\frac{1}{\left(\frac{\lambda-\Delta}{4}\right)^2\left(\frac{1-\left(\frac{\lambda+\Delta}{\lambda-\Delta}\right)^{n+2}}{1-\left(\frac{\lambda+\Delta}{\lambda-\Delta}\right)^{n+1}}\right)-\left(\frac{1-\left(\frac{\lambda+\Delta}{\lambda-\Delta}\right)^{n+1}}{1-\left(\frac{\lambda+\Delta}{\lambda-\Delta}\right)^{n+2}}\right)} \tag{22}$$

Using Claim B.2 point 4. to bound the ratio

$$\frac{1-\left(\frac{\lambda+\Delta}{\lambda-\Delta}\right)^{n+2}}{1-\left(\frac{\lambda+\Delta}{\lambda-\Delta}\right)^{n+1}}\geq 1+\frac{1}{2(n+1)},$$

the denominator of (22) is at least

$$\left(\frac{\lambda-\Delta}{4}\right)^2\left(\frac{1-\left(\frac{\lambda+\Delta}{\lambda-\Delta}\right)^{n+2}}{1-\left(\frac{\lambda+\Delta}{\lambda-\Delta}\right)^{n+1}}\right)-\left(\frac{1-\left(\frac{\lambda+\Delta}{\lambda-\Delta}\right)^{n+1}}{1-\left(\frac{\lambda+\Delta}{\lambda-\Delta}\right)^{n+2}}\right)\geq\left(\frac{\lambda-\Delta}{4}\right)^2\left(1+\frac{1}{2(n+1)}\right)-1. \tag{23}$$

We now observe that

$$\left(\frac{\lambda-\Delta}{4}\right)^2=\frac{\lambda^2+\Delta^2-2\lambda\Delta}{16}$$
$$=\frac{2(8+2\sqrt{8}\gamma+\gamma^2)-8-2\lambda\Delta}{16}$$
$$\geq\frac{1}{2}+\frac{4\sqrt{8}\gamma+2\gamma^2-24\sqrt{\gamma}}{16}$$
$$>\frac{1}{2}-\sqrt{\gamma}$$
$$\geq\frac{1}{2}-\frac{1}{16(n+2)}.$$

Continuing from (23), we have that the denominator of (22) is at least

$$\left(\frac{1}{2}-\frac{1}{16(n+2)}\right)\left(1+\frac{1}{2(n+1)}\right)-1\geq-\frac{1}{2}+\frac{1}{4(n+1)}-\frac{1}{16(n+1)}-\frac{1}{32(n+1)^2}\geq-\frac{1}{2}+\frac{1}{8(n+1)}.$$

We may also upper bound the denominator as

$$
\left(\frac{\lambda-\Delta}{4}\right)^2 \left(\frac{1-\left(\frac{\lambda+\Delta}{\lambda-\Delta}\right)^{n+2}}{1-\left(\frac{\lambda+\Delta}{\lambda-\Delta}\right)^{n+1}}\right) - \left(\frac{1-\left(\frac{\lambda+\Delta}{\lambda-\Delta}\right)^{n+1}}{1-\left(\frac{\lambda+\Delta}{\lambda-\Delta}\right)^{n+2}}\right) \leq
$$

$$
\left(\frac{\lambda-\Delta}{4}\right)^2 - \left(\frac{1-\left(\frac{\lambda+\Delta}{\lambda-\Delta}\right)^{n+1}}{1-\left(\frac{\lambda+\Delta}{\lambda-\Delta}\right)^{n+2}}\right) \leq
$$

$$
\frac{1}{2} - \left(1 + \frac{1}{2(n+1)}\right)^{-1} =
$$

$$
\frac{1}{2} - \frac{2(n+1)}{2(n+1)+1} =
$$

$$
\frac{1}{2} - 1 + \frac{1}{2(n+1)+1} \leq
$$

$$
-\frac{1}{2} + \frac{1}{2(n+1)}.
$$

Using these bounds in (22) we conclude

$$
\alpha\left(\left(\frac{\lambda-\Delta}{\lambda+\Delta}\right)^{n+1} - 1\right) \leq \frac{1}{-1 + \frac{1}{4(n+1)}}
$$

$$
= -\left(\frac{1}{1 - \frac{1}{4(n+1)}}\right)
$$

$$
= -\left(1 + \frac{\frac{1}{4(n+1)}}{1 - \frac{1}{4(n+1)}}\right)
$$

$$
\leq -1 - \frac{1}{4(n+1)}.
$$

and

$$
\alpha\left(\left(\frac{\lambda-\Delta}{\lambda+\Delta}\right)^{n+1} - 1\right) \geq \frac{1}{-1 + \frac{1}{n+1}}
$$

$$
= -\left(\frac{1}{1 - \frac{1}{n+1}}\right)
$$

$$
= -\left(1 + \frac{\frac{1}{n+1}}{1 - \frac{1}{n+1}}\right)
$$

$$
\geq -1 - \frac{2}{n+1}.
$$

$\square$

