# OpenReview forum: "An Exponential Separation Between Quantum and Quantum-Inspired Classical Algorithms for Linear Systems"
_ICML.cc/2026/Conference — ICML 2026 regular_

### Official Review · Reviewer_CWBp · 2026-02-26

**Soundness:** 3
**Presentation:** 2
**Significance:** 2
**Originality:** 3
**Overall Recommendation:** 2
**Confidence:** 4

**Summary:**

The work seems to present the first such provable exponential separation between quantum and quantum-inspired classical algorithms for the basic problem of solving a linear system when the input matrix is well-conditioned and has sparse rows and columns.

**Compliance With Llm Reviewing Policy:**

Affirmed.

**Final Justification:**

I maintain my score.

**Key Questions For Authors:**

None.

**Limitations:**

Out of scope.

**Strengths And Weaknesses:**

This submission presents results that would likely be of strong interest to the theoretical computer science community, particularly in complexity theory. The paper develops complexity-theoretic bounds, relating highly efficient quantum and quantum-inspired classical algorithms for the solution of small family of linear systems. The technical contributions appear solid and well-aligned with venues such as ICALP, MFCS or STOC.

However, in its current form, the paper seems out of scope for ICML. The work does not include experiments, nor does it develop theoretical insights that are directly grounded in machine learning models, learning guarantees, or empirical practice. While the motivation is that linear system solvers are important for machine learning, the connection remains quite indirect. It is unclear what concrete insight for ML theory or practice is gained from establishing the presented complexity result. In particular, the algorithms considered rely on sparsity and strong assumptions on the condition number that are violated by the vast majority of matrices arising in modern machine learning applications, which further limits the practical relevance of the results to the ICML audience.

Overall, this appears to be a well-executed complexity-theoretic contribution, but one that would be better suited for a conference in complexity theory rather than a machine learning venue such as ICML.

---

> ### Author Rebuttal · Authors · 2026-03-30
>
> We are sorry that the reviewer feels the paper is out of scope. We respectfully disagree and find that a solid theoretical understanding of what benefits quantum might bring to ML is an important goal.
>
> That said, let us also clarify one point of confusion. The reviewer says that the lower bound "rely on sparsity and condition number", making it unclear what relevance it has. This is the wrong way around. If we were designing an algorithm, then having it rely on sparsity and condition number would be bad as it restricts the applicability of the algorithm. Since we are proving a lower bound, the lower bound is STRONGER. It says, even if you are willing to restrict yourself to matrices that are sparse and have good conditioned number, then you can never hope to design an algorithm that is efficient in the quantum inspired classical setting. At least not without EVEN MORE assumptions than sparsity and good condition number.

---

> > ### Author Rebuttal · Reviewer_CWBp · 2026-04-03
> >
> > Thanks for the technical clarification which essentially says that "EVEN MORE assumptions than sparsity and good condition number" would be required to make use of HHL-like algorithms in the solution of linear systems. Not sure this makes the result more relevant for ML though. At the end, this paper is imho out of scope.

---

### Official Review · Reviewer_HXEG · 2026-03-07

**Soundness:** 4
**Presentation:** 3
**Significance:** 3
**Originality:** 3
**Overall Recommendation:** 5
**Confidence:** 4

**Summary:**

The paper studies the canonical problem of solving a linear system Mx=y, and proves a classical lower bound that, in a certain parameter regime, is exponentially higher than an existing quantum upper bound. Combined with a known quantum algorithm (Charkraborty et al 2019, a follow-up work of HHL), this shows an exponential gap between classical and quantum query complexity of a natural problem.
(Caveats:
1. The solution is not to find x but sampling i roughly with probability |x_i|^2 / ||x||^2.
2. The gap is large only in certain parameter regimes.)

The paper gives two lower bounds by reducing the problem to two different problems, Glued Tree and Forrelation, for which strong classical lower bounds were known. The main text only proves the first one, including both oracle and algorithm transformations.

**Compliance With Llm Reviewing Policy:**

Affirmed.

**Final Justification:**

The result provides the first exponential separation between quantum algorithms and the “quantum-inspired-classical algorithms“, which is theoretically novel and interesting. Such lower bounds are intellectually exciting, and useful in that no algorithms along this direction can do better (and thus one should not try to improve existing algorithms along this line!).  I keep my score at 5.

**Key Questions For Authors:**

On the more practical side, while the problem of linear solver is natural, the hard instances in the proof showing the lower bounds are quite artificial. It'd be great (even for the theory community) if the authors can also identify more practically relevant instances with large classical-quantum gaps.

**Limitations:**

Yes

**Strengths And Weaknesses:**

Strengths
1. The result provides the first exponential separation between quantum algorithms and the “quantum-inspired-classical algorithms“, which is theoretically novel and interesting.

2. The proof is straightforward if the design of matrix M is given, but the design itself is nontrivial. The authors have made nice efforts to explain the intuitions behind the parameter design.

Weaknesses
As a theoretical paper this paper can stand on its own, with no apparent weaknesses.

---

> ### Author Rebuttal · Authors · 2026-03-30
>
> Thank you for your review! And yes, identifying more classes where the separation holds is a worthwhile undertaking. At least our lower bound points to the fact that any quantum inspired classical algorithm that one would want to prove efficient for a class of matrices, has to take more parameters into account than sparsity and condition number.

---

> > ### Author Rebuttal · Reviewer_HXEG · 2026-04-01
> >
> > These lower bound results are novel and significant theoretically in the quantum learning area. I maintain my score 5.

---

### Official Review · Reviewer_nTxz · 2026-03-11

**Soundness:** 3
**Presentation:** 3
**Significance:** 4
**Originality:** 4
**Overall Recommendation:** 4
**Confidence:** 3

**Summary:**

This paper asks whether quantum algorithms for linear systems still have a true exponential advantage once they are compared against the stronger “quantum-inspired” classical model that gets the same sampling-style access to the input. The main contribution is an unconditional exponential separation: the authors construct sparse, well-conditioned linear systems for which any quantum-inspired classical algorithm needs polynomially many queries in n, while known quantum algorithms solve the same task with only polylogarithmic overhead. Technically, they prove this through lower bounds based on reductions from hard classical query problems, with two alternative proof routes via random walks on binary trees and via k-Forrelation. Overall, the paper positions linear-system solving as one of the first natural machine-learning-relevant problems where an exponential quantum advantage survives against the quantum-inspired classical framework.

**Compliance With Llm Reviewing Policy:**

Affirmed.

**Key Questions For Authors:**

1) How “natural” is the constructed family of linear systems, beyond satisfying sparsity and conditioning constraints?
The paper convincingly proves a separation for a carefully designed instance family, but I would like the authors to comment more explicitly on whether the construction captures broader classes of linear systems that arise in quantum ML or numerical linear algebra, or whether it should mainly be viewed as a complexity-theoretic witness. A strong answer here would increase my view of the paper’s significance, especially for an ML audience.

2) Can the authors clarify which parts of the argument fundamentally require the specific SQ-access/QIC model, and which parts might extend to stronger classical access models?
Since the paper’s message is partly about the limits of dequantization, it would help to understand how robust the separation is to reasonable strengthening of the classical model. If the authors can argue that the separation survives in a broader setting, that would strengthen my assessment of both significance and originality.

**Limitations:**

The paper includes an impact statement, but it is too brief to count as an adequate discussion of limitations.
A stronger version would explicitly note that this is a foundational theory paper with limited immediate deployment risks, while still acknowledging possible longer-term effects such as influencing how researchers and funders assess claims of quantum advantage in ML. It would also help to state the main limitations more clearly: the separation is shown for carefully constructed instance families in a particular access model, and it does not by itself establish practical quantum advantage for real-world linear systems or current ML applications.

**Strengths And Weaknesses:**

The paper appears technically strong: its main claim is stated precisely, and the proof strategy is well matched to the result, using reductions from problems with strong classical lower bounds. The presence of two complementary proof routes increases confidence, though the argument is intricate enough that full verification still requires careful appendix-level checking.
It is generally clear and well organized, with a strong introduction that explains the motivation and places the work in the dequantization literature. Its main weakness is density: the proofs are technically heavy, and the paper would benefit from a more explicit roadmap and a clearer comparison between the two reductions.
This paper addresses an important question in quantum machine learning theory: whether exponential quantum advantages survive against quantum-inspired classical algorithms. Its contribution is mainly foundational rather than practical, but it meaningfully advances understanding of where dequantization breaks down and is likely to influence future theoretical work.
The work is highly original in both result and technique: it establishes a new exponential separation for linear-system solving in the quantum-inspired setting. Even though it builds on existing lower-bound tools, the way those ideas are combined and adapted to this problem is creative and clearly novel.

---

> ### Author Rebuttal · Authors · 2026-03-30
>
> Thank you for your review.
>
> For the "how natural" question. We would like to point out that since we are proving a lower bound, not developing a new algorithm, proving the lower bound EVEN under the restrictions of sparsity and well conditioned only makes the lower bound stronger. It implies that if we want to design an algorithm and prove that it is efficient, we need to place even more assumptions and restrictions on the type of input matrix if we are to hope of proving a fast upper bound. That said, one can also view our lower bound in a positive light: It points at concrete obstacles that any fast algorithm has to work around. In particular, any fast algorithm has to make or identify additional assumptions about real data.
>
> Regarding the exact model, note our comment on the beginning of the very last paragraph on page 3: The lower bound also holds in the standard sparse access model. We are not aware of other query models that adequately takes preprocessing of the matrix into account and in which it would be natural to ask whether a similar lower bound holds.

---

> > ### Author Rebuttal · Reviewer_nTxz · 2026-04-01
> >
> > My concerns about the naturalness of the assumptions and the query model are adequately addressed. The rebuttal clarifies that, since the paper proves a lower bound, showing hardness even under sparsity and good conditioning strengthens the result.

---

### Official Review · Reviewer_zRv3 · 2026-03-11

**Soundness:** 2
**Presentation:** 1
**Significance:** 1
**Originality:** 2
**Overall Recommendation:** 3
**Confidence:** 2

**Summary:**

This work proves an (unconditional) exponential separation between quantum and quantum-inspired classical (QIC) algorithms for solving sparse linear systems. The "hard instance matrix" $M = \lambda I - A$ is constructed using two-binary-tree graph from the Childs et al. (2003), with $\lambda = \sqrt{8} + O(1/n^2)$. It can then be shown that any QIC algorithm needs $\Omega(n^{1-1/k})$ queries to $SQ(M)$ while quantum algorithm solve the same task in $\text{poly}(\ln n)$ time. The key technical ingredients involve: (i) a reduction showing each $SQ(M)$ query can be simulated with $O(1)$ oracle query, and (ii) a spectral analysis proving the matrix is well-conditioned and the root coordinate satisfies certain technical conditions.

**Compliance With Llm Reviewing Policy:**

Affirmed.

**Final Justification:**

I appreciate the authors' response, and have raised the score to 3. Unfortunately I still believe ICML is not the right venue for this work. Reviewer CWBp seems to agree.

**Key Questions For Authors:**

- The hard instance relies on quite specific structure: two binary trees with randomly joined leaves. Is there any intuition/characteristic of $M$ that is essential for the separation? Is it possible to understand a "family" of matrices where such unconditional exponential separation holds?

**Limitations:**

Yes

**Strengths And Weaknesses:**

### Strengths
* This is the first unconditional exponential separation between quantum and QIC algorithms for the fundamental problem of solving linear systems of equations. Previous separation results were either conditional under complexity-theoretic assumptions (BQP $\neq$ P), or only showed polynomial separation.
* Two independent reductions (graph random walk and $k$-Forrelation) are provided, complementing each other.
* The hard instance is considered has "clean" properties: 3-sparse, symmetric, real, and well-conditioned.

### Weaknesses
* It is not clear to me if venues like ICML is a good fit for this work. A bulk of the main text reads as a survey/review of QIC algorithms, HHL, and prior lower bounds. Furthermore, to the quantum algorithms/complexity community, the exponential separation in the sparse access case is known. While it's true that the previous separations were "conditional" and proving such unconditional separation (as this work shows) is nontrivial, the result seem too specific/niche for ML community.
* Moreover, the hard instance is highly structured (two binary trees + random cycle on leaves). Hence, it is unclear whether the lower bound extends to broader, more natural classes of linear systems encountered in practice, which would be of bigger interest to ML community.
* In my humble opinion, this work fits in quantum/TCS community better than ML community.

---

> ### Author Rebuttal · Authors · 2026-03-30
>
> Thank you for your review. We are sorry that you feel this is not an appropriate fit for ICML. We respectfully disagree and feel a better understanding of what quantum might hope to bring to ML is a fundamental theoretical goal.
>
> Regarding which types of matrices might be hard. It is indeed a bit unclear. The two examples so far both rely on a problem that is hard in an oracle setting. While it is unclear whether these matrices correspond to practically occurring matrices, we believe the lower bound still has value. In particular, if one would try to design a new quantum inspired classical algorithm, then our lower bound says that any analysis showing that the algorithm is efficient HAS TO take into account additional parameters other than sparsity and condition number. In that sense, one can alternatively view the lower bound in a positive light, as pointing towards obstacles that any upper bound has to circumvent.

---

> > ### Author Rebuttal · Reviewer_zRv3 · 2026-04-03
> >
> > I appreciate the authors' response. but I still believe ICML is not the right venue for this work. Reviewer CWBp seems to agree.

---

### Decision · Program_Chairs · 2026-04-30

**Decision:**

Accept (regular)

**Comment:**

After thinking about the reviews, I am in favor of accepting this paper. Quantum-inspired machine learning has received a lot of attention in ML conferences such as ICML (e.g., https://proceedings.mlr.press/v162/chepurko22a.html ). Yes, this paper is perhaps less practically relevant as the lower bounds are somewhat artificial and parameter regimes where the separation hold are not so broad, but I view it as a very important first step in this direction, and a result that has eluded the area.